

# WAVETRISK-2.1: an adaptive dynamical core for ocean modelling

Nicholas K.-R. Kevlahan[1] and Florian Lemarié[2]

[1]Department of Mathematics and Statistics, McMaster University, Hamilton, Canada
[2]Université Grenoble Alpes, Inria, CNRS, Grenoble INP, LJK, Grenoble, France

**Correspondence:** N. K.-R. Kevlahan (kevlahan@mcmaster.ca)

**Abstract.** This paper introduces WAVETRISK-2.1 (i.e. WAVETRISK-OCEAN), an incompressible version of the atmosphere model WAVETRISK-1.x with free-surface. This new model is built on the same wavelet-based dynamically adaptive core as WAVETRISK, which itself uses DYNAMICO's mimetic vector-invariant multilayer rotating shallow water formulation. Both codes use a Lagrangian vertical coordinate with conservative remapping. The ocean variant solves the incompressible multi-layer shallow water equations with inhomogeneous density layers. Time integration uses barotropic–baroclinic mode splitting via an semi-implicit free surface formulation, which is about 34–44 times faster than an unsplit explicit time-stepping. The barotropic and baroclinic estimates of the free surface are reconciled at each time step using layer dilation. No slip boundary conditions at coastlines are approximated using volume penalization. The vertical eddy viscosity and diffusivity coefficients are computed from a closure model based on turbulent kinetic energy (TKE). Results are presented for a standard set of ocean model test cases adapted to the sphere (seamount, upwelling and baroclinic turbulence). An innovative feature of WAVETRISK-OCEAN is that it could be coupled easily to the WAVETRISK atmosphere model, thus providing a first building block toward an integrated Earth-system model using a consistent modelling framework with dynamic mesh adaptivity and mimetic properties.

## 1 Introduction

Dynamically adaptive methods have the potential to significantly improve the computational efficiency and accuracy of the dynamical cores of atmosphere and ocean models. They do this by optimizing grid resolution at each time step to represent the dynamically active parts of the flow. This makes better use of computational resources by using fine resolution where needed, and also allows better control of accuracy since the grid may be adapted based on a local error indicator. The same technique can also be used to build statically adapted "nested" models which avoid reflection and other errors at the refinement boundaries. Another feature of adaptive methods is that they can be run at coarse resolutions for long times to spin up a model, and then easily restarted with much higher resolutions for shorter runs.

However, these advantages come at the cost of increased code complexity and it is not clear *a priori* whether dynamically adaptive methods will work well in complex multi-physics simulations with separate subgrid scale (SGS) parameterizations. Because of their potential, we have been pursuing a program to push the adaptive paradigm as far as possible, to help assess its potential in realistic or semi-realistic Earth system models. Our approach uses the powerful wavelet collocation multiresolution framework, adapted to the needs of geophysical fluid dynamics (Kevlahan, 2021). We began with the shallow water equations on the $\beta-$plane (Dubos and Kevlahan, 2013) and extended this method to the sphere (Aechtner et al., 2015). Ocean models





require accurate approximation of boundary conditions at solid boundaries, and Kevlahan et al. (2015) derived a robust volume penalization scheme to implement no-slip boundary conditions in an adaptive model. Debreu et al. (2020) extended the volume penalization approach to modelling ocean bathymetry. Finally, Kevlahan and Dubos (2019) extended this two-dimensional

models to a three-dimensions hydrostatic atmosphere model using Lagrangian vertical coordinates.

To simplify development, and for compatibility with existing SGS parameterizations, adaptivity is horizontal only (as in Popinet (2021)). This means the data structure is a set of vertical columns of varying resolutions. WAVETRISK is parallelized using mpi, and exhibit good strong parallel scaling properties (Kevlahan and Dubos, 2019).

This paper presents the final stage in the development of a foundational set of dynamical cores for adaptive Earth system

models. WAVETRISK-2.1 (which we will refer to as WAVETRISK-OCEAN) is a three-dimensional hydrostatic free-surface ocean model with Lagrangian vertical coordinates and inhomogeneous density layers.

In the terminology of Beron-Vera (2021) WAVETRISK-OCEAN is an $n$-IL$^0$ model, i.e. an inhomogenous-layer model where variables do not vary vertically within each layer. This is in contrast to the more common homogeneous layer ($n$-HL) models, where buoyancy is horizontally and vertically homogeneous in each layer. The single layer IL$^0$ model was introduced by Ripa

(1993) to represent thermodynamic processes in a single layer, and is sometimes called a "thermal rotating shallow-water model". $n$-IL$^0$ preserves important mimetic properties of the continuously stratified system (Kelvin's circulation theorem, advection of potential vorticity, conservation of Casimir invariants) and ensures a good approximation of the horizontal pressure gradient (similar to state-of-the-art models using terrain following coordinates). Beron-Vera (2021) improves $n$-IL$^0$ to $n$-IL$^1$ by allowing linear vertical variation within each layer.

WAVETRISK-OCEAN includes barotropic-baroclinic mode splitting using a semi-implicit free surface method implemented using a $\theta$-method in time. The associated linear elliptic problem is solved efficiently using an adaptive multigrid method based on the multiscale wavelet grid structure. According to the classification proposed by Griffies et al. (2020) WAVETRISK-OCEAN is based on a vertical Lagrangian-remap method, as illustrated in their Figure 3.

The current version of WAVETRISK-OCEAN is semi-realistic, since it includes some basic features of a practical ocean model.

These include conservative grid remapping, inclusion of complex coastline geometries and bathymetry, and vertical diffusion using a turbulent kinetic energy (TKE) closure scheme. Solar flux and wind stress forcing are also options. Where possible, we have tried to incorporate best practice features of well-established ocean models, such as NEMO (Madec and Team, 2015) and MITGCM (Adcroft et al., 2021). This model is sufficiently realistic to serve as a test bed for adaptive modelling of ocean flows, while avoiding the complexity of a true operational ocean model.

Popinet (2021) has recently introduced an adaptive regional non-hydrostatic/hydrostatic multilayer ocean model built on the Basilisk framework that he had used previously for two-dimensional shallow water ocean modelling (Popinet and Rickard, 2007; Popinet, 2011). This model uses Lagrangian layers and the same remapping we use here (Engwirda and Kelley, 2016). As in WAVETRISK, he adapts the grid horizontally, but not vertically. However, in contrast with WAVETRISK-OCEAN model, Popinet (2021) does not use barotropic–baroclinic mode splitting, does not use penalization for solid boundaries, is based on

a fundamentally different adaptivity method, and is a regional rather than global model. WAVETRISK-OCEAN also includes a vertical mixing parameterization, which is essential for climate modelling. Popinet (2021)'s model has been developed for





short time scale regional simulations, while WAVETRISK-OCEAN is aimed at global climate modelling of the oceans. Finally, an important design goal of WAVETRISK-OCEAN was to incorporate important mimetic properties in the adaptive discretization. These two adaptive models are therefore complementary, and provide a good illustration of the applicability of adaptivity to ocean modelling.

The dynamical equations and basic approximations of the model are summarized in Section 2 and the components of the numerical scheme are described in detail in Section 3. Results for a set of ocean model tests cases are presented in Section 4. We summarize our main conclusions and outline some perspectives for future use and development of WAVETRISK-OCEAN in Section 5.

## 2 Dynamical equations and adaptivity

### 2.1 Dynamical equations

This initial release of WAVETRISK-OCEAN uses the incompressible version of the DYNAMICO (Dubos et al., 2015) equations on an icosahedral C-grid with Lagrangian vertical coordinates. These exactly incompressible equations are based on the *simple Boussinesq* approximation (Vallis, 2006), which neglects the hydrostatic compressibility of seawater. This means that the thermodynamic equation is based on density (i.e. buoyancy) and not on potential density,

$$\rho_{\text{pot}} = \rho - \frac{\rho_0 g z}{c_s^2} \approx \rho \qquad (1)$$

where $c_s \approx 1500\,\mathrm{m/s}$ is the speed of acoustic waves. This choice ensures a consistent and mathematically well-founded approximation of the Navier–Stokes equations based on Hamilton's principle and the associated Euler–Lagrange equations (Dubos et al., 2015) at the cost of some loss of realism. The test cases presented here all use a simple linear equation of state relating density and temperature

$$\rho = \rho_0 + a_0(T - T_a), \qquad (2)$$

where linear coefficient of thermal expansion $a_0 \approx 0.1655\,\mathrm{kg/meter}^3/°\mathrm{C}$, and the reference temperature $T_a \approx 10\,°\mathrm{C}$ (actual values depend on the test case).

For simplicity, we present the equations in non-penalized form (i.e. with open boundary conditions). In Section 3.4 we review briefly the volume penalization used to approximate solid boundaries (e.g. continents) originally developed in Kevlahan et al. (2015).

The prognostic variables are inertial pseudo-density $\mu_{ik} = \rho_0 \Delta z_{ik}$ (using the Boussinesq approximation), mass-weighted buoyancy, $\Theta_{ik} = \mu_{ik}\theta_{ik}$ (where we define buoyancy as $\theta_{ik} = 1 - \rho_{ik}/\rho_0$) and velocity $v_{ek}$. Index $k$ labels a full vertical layer, $l$ an interface (half-layer) between full vertical layer, $i$ an hexagonal or pentagonal cell, $v$ a triangle and $e$ an edge, with geometry as shown in figure 1. The dynamical equations of motions (without splitting the barotropic and baroclinic modes) are

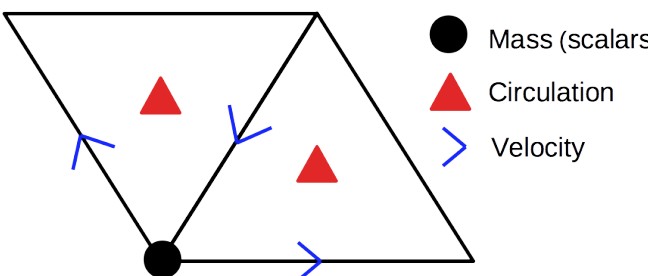

**Figure 1.** Basic computational cell for the icosahedral C-grid discretization, containing one node (for mass and buoyancy), three edges (for velocities) and two triangles (for circulation). Separate wavelet transforms are provided for the nodes (scalar-valued) and edges (vector-valued). The adaptive grid consists of the the significant nodes and edges, together with nearest neighbours in position and scale necessary for dynamics. The horizontal grid is the same in each vertical layer.

$$\partial_t \mu_{ik} + \delta_i F_{ek} = 0, \tag{3}$$

$$\partial_t \Theta_{ik} + \delta_i(\overline{\theta_k}^e F_{ek}) = D_\phi \Theta_{ik}, \tag{4}$$

$$\partial_t v_{ek} + \delta_e B_{ik} - \overline{\theta_k}^e \delta_e \overline{\Phi_i}^k + (q_k F_k)_e^\perp = D_\delta v_{ek} + D_\omega v_{ek}, \tag{5}$$

where $F_{ek} = \overline{\mu_k}^e v_{ek}$ is the horizontal mass flux, $q_{ek}$ is the potential vorticity, and we have assumed Lagrangian vertical co-ordinates (so the vertical mass fluxes are not explicit). Centred averages are used for all interpolated quantities, e.g. $\overline{(\cdot)}^e$ is a node quantity reconstructed at an edge. The discrete operators $\delta_i$ (divergence, with result at a node), $\delta_e$ (gradient, with result at an edge) and $(\cdot)^\perp$ (perpendicular flux) are defined as in Ringler et al. (2010). Note that the top vertical layer $k = N$ includes the free surface perturbation with the interface $l = N + 1$ at the free surface, and the bottom vertical interface $l = 1$ is the bathymetry. We include an additional $N + 1$ vertical layer to represent the separate free surface variable $\eta$ when splitting the baroclinic and barotropic modes. In the examples we consider here we use a hybrid $\sigma - z$ grid. In the upwelling and baroclinic jet cases the hybrid grid layers are thinner near the free surface, using a profile similar to that described in Shchepetkin and McWilliams (2009).

The system (3-5) is a multi-layer rotating shallow water model with inhomogeneous density layers (i.e. $\delta\theta_k \neq 0$), but assumes zero vertical variation of velocity and buoyancy within each layer (i.e. an $n-\text{IL}^0$ model). A similar model was derived in flux-form by Ripa (1993) and in vector-invariant form by Dubos et al. (2015). In this model, to be consistent with the piecewise constant representation of $v$ and $\theta$ in the vertical, a vertical average of the horizontal pressure gradient term in each layer is used to compute horizontal velocity (Ripa, 1993).

The Bernoulli function for hydrostatic incompressible flow is

$$B_{ik} = K_{ik} + \overline{\Phi_i}^k + \frac{\lambda_{ik}}{\rho_0}, \tag{6}$$

where $K_{ik}$ is the discrete kinetic energy computed from $v_{ek}$ using appropriate averaging, and $\Phi_{il}$ is the geopotential at vertical layer interfaces $l$. Pressure $\lambda_{ik}$ is calculated by summing the hydrostatic contribution from each vertical layer, $g(1 - \theta_{ik})\mu_{ik}$,





from the top down. The hydrostatic pressure is therefore given by

$$\lambda_{ik} = \sum_{j=k}^{N} g(1 - \theta_{ij})\mu_{ij} - \frac{1}{2}g(1 - \theta_{ik})\mu_{ik} = \sum_{j=k+1}^{N} g(1 - \theta_{ij})\mu_{ij} + \frac{1}{2}g(1 - \theta_{ik})\mu_{ik}.$$

The terms on the right hand side of (3–5) are the discretizations of the appropriate Laplacian horizontal diffusion operators, $D_\phi = \nabla \cdot (K_\phi \nabla \phi)$ (for the scalars) and $D_\delta = \nabla(K_\delta \nabla \cdot v)$ and $D_\omega = \nabla \times (K_\omega \nabla \times v)$ (for the velocity),

$$D_\phi = \delta_i \left[ K_\phi \frac{l_e}{d_e} \delta_e \left[ \frac{\phi}{A_i} \right] \right], \tag{7}$$

$$D_\delta = \delta_e \left[ K_\delta \frac{1}{A_i} \delta_i \left[ \frac{l_e}{l_d} v_e \right] \right], \tag{8}$$

$$D_\omega = \delta_e \left[ K_\omega \frac{1}{A_v} \delta_v [v_e] \right], \tag{9}$$

where $d_e$ is a triangle edge length (primal grid), $l_e$ is a hexagon edge length (dual grid), $A_i$ is a hexagon area, $A_v$ is a triangle area. The horizontal diffusion coefficients $K_\phi$, $K_\delta$ and $K_\omega$ are constants, and can be chosen either to model physical diffusion, or at minimal values to ensure stability. In general $K_\phi = 0$, although some grid scale horizontal diffusion on the Lagrangian layer thicknesses $\mu_{ik} = \rho_0 \Delta z_{ik}$ and buoyancy could be included on the right hand side of equation (3) to enhance numerical stability. For better accuracy and stability, mass density (i.e. layer depth) is decomposed into its mean and fluctuating parts and
we solve for the fluctuations.

## 2.2 Vertical remapping and horizontal grid adaptivity

Prognostic variables may be remapped as desired onto a target vertical grid using a conservative piecewise parabolic remapping scheme, as described in Kevlahan and Dubos (2019), to avoid layer collapse or to ensure desired properties of the vertical grid (e.g. approximately isopycnal).

The horizontal grid is adapted on fluctuating pseudo density (i.e. perturbations from mean layer depths), mass-weighted buoyancy and velocities. Adapting on pseudo density ensures that the deformations of the Lagrangian layer interfaces are properly represented by the adaptive grid. Note that if buoyancy is initially constant in each layer, i.e. $\theta_{ik} = \overline{\theta}_k$, and the vertical grids are not remapped, $\Theta_{ik} = (\overline{\mu}_{ik} + \mu_{ik})\overline{\theta}_k$, i.e. buoyancy remains constant in each layer.

The horizontal grid adaptation scheme is based on the fact that wavelet coefficients measure the interpolation error at each
position and scale. A unique grid point is associated to each wavelet and so removing (small) wavelets from the data structure also removes the corresponding grid point, resulting in an adapted grid.

The essentials of the horizontal grid adaptation strategy are as follows. At the end of a time step the wavelet coefficients of the prognostic variables are computed separately for each horizontal layer. Wavelet coefficients larger than the specified relative tolerance $\varepsilon$ for each prognostic variable are retained, and the remainder are deleted. This produces a multiscale adapted
grid for each vertical layer. The actual adapted grid is then the union of the adapted grids over all vertical layers. To account for the change in the solution over one time step, nearest neighbours are then added in both scale and position. This is sufficient for a dynamical equation with a quadratic nonlinearity and a time step corresponding to an advective CFL criterion of one.





Additional points are added to ensure that the adapted grid includes the stencils required for all discrete differential operators. Finally, all variables are inverse wavelet transformed onto the new adapted grid.

The resulting adapted horizontal grid is the same in each vertical layer, which means that the computational elements are a collection of columns of various sizes at each level of resolution $j$. Full details of the horizontal grid adaptation algorithm are available in Kevlahan and Dubos (2019); Kevlahan (2021).

In the incompressible version of WAVETRISK described in Kevlahan and Dubos (2019) the grid is adapted after each time step, since the time step is based on the advective CFL number. However, in WAVETRISK-OCEAN the time step is usually

significantly smaller than the advective time step, since the advective velocity $U \approx 1 \, \text{m/s}$ is much smaller than the barotropic velocity $U \approx 200 \, \text{m/s}$. This means that, even in the mode split version (3.1), the grid can be adapted much less frequently, leading to a cpu time saving of about 10% per time step. For example, in the unstable baroclinic jet case (§4.3), which uses a barotropic CFL criterion of 35, the grid can be adapted every 8 time steps. This strategy is based on the fact that high resolution is needed primarily to track the fine scale vorticity filaments and associated density/temperature fluctuations (i.e. the turbulent

geostrophic modes).

We use a single time step for all resolution levels $j$. This may be less efficient than using a resolution-dependent time step in cases where a majority of active grid points are at the finest levels (i.e. low levels of adaptivity), but it greatly simplifies the time stepping algorithm, especially in the mode split case. We may consider implementing a resolution-dependent Runge–Kutta method (McCorquodale et al., 2015) in future versions of WAVETRISK.

## 3   Numerical scheme

### 3.1   Barotropic–baroclinic mode splitting time step

The barotropic (or external) mode is typically $O(10^2)$ faster than the baroclinic (internal gravity wave) modes and advective time scales of the flow. For an ocean of mean depth $H = 4 \, \text{km}$ the external wave speed is approximately $c_0 = \sqrt{gH} \approx 200 \, \text{m/s}$, while the typical advective velocity is $U \approx 1 \, \text{m/s}$ (the first baroclinic mode is usually much slower, typically

$c_1 = \frac{c_0}{\pi} \sqrt{-\frac{H}{\rho_0} \frac{d\rho}{dz}} \approx 0.05 \, \text{m/s}$). To avoid advancing all vertical layers at the very small time step set by the stability criterion for the external modes, most ocean models solve separately the two-dimensional barotropic mode and the three-dimensional baroclinic modes. This may be done using explicit sub-cycling (e.g. ROMS, Shchepetkin and McWilliams (2005) and MPAS-O Kang et al. (2021)), by taking small time steps $\Delta t \sim \Delta x / c_0$ for the two-dimensional barotropic mode and longer time steps $\Delta t \sim \Delta x / U$ for the baroclinic modes (e.g. NEMO Madec and Team, 2015), imposing a "rigid lid" (no longer used in operational

models), or with implicit time stepping for the free surface. The implicit free surface filters the fast unresolved wave motions by damping them, and does not require an extremely accurate solution of the associated elliptic equation (unlike the rigid lid approach).

The implicit free surface approach has been used in established ocean models such as MITGCM (Marshall et al., 1997; Adcroft et al., 2021), as well as in more recent ocean models such as FESOM (Danilov et al., 2017) and MPAS-O (Kang et al.,

2021). Implicit free surface models are a natural choice for unstructured grid models with variable resolution.





WAVETRISK-OCEAN allows two time stepping schemes: explicit low-storage RK4 without mode splitting, and barotropic–baroclinic mode splitting with a linear implicit free surface. The fully implicit free surface method is unconditionally stable for the barotropic mode, although stability requirements for the baroclinic and vortical modes limit the practically useful barotropic CFL number. Since the implicit time stepping scheme is strongly diffusive, the computed free surface waves are

strongly diffused at large values of $C_{\text{barotropic}}$. Therefore, fully implicit mode splitting is appropriate only when we are interested primarily in the slow baroclinic dynamics. The following linear free surface scheme shares some features of the barotropic–baroclinic $\theta$-step used in MITGCM (e.g. Adcroft et al., 2021, section 2.4).

Because of the significant dissipation associated with the fully implicit method, we implement a $\theta$ semi-implicit time integration method, where the parameter $1/2 \leq \theta \leq 1$ determines the mix of implicit and explicit approximations of the barotropic

flow divergence and surface pressure gradient components. $\theta = 1$ gives the full implicit scheme, while $\theta = 1/2$ gives a Crank–Nicolson scheme (non-dissipative, but less stable). We will see that when using RK3 or RK4 for the explicit part, the $\theta$ method is unconditionally stable for $\theta \geq 0.75$. For simplicity, we describe in detail only the fully implicit $\theta = 1$ method, using explicit Euler. The general $\theta$-method is a simple modification, implemented as in MITGCM (Adcroft et al., 2021, section 2.10.1). Note, that the explicit Euler method is unconditionally unstable, and the actual implementation uses third or fourth order Runge–

Kutta, which are unconditionally stable for $\theta \geq 0.75$. The stability properties of the time integration scheme is discussed at the end of this section.

Consider a first order discretization of the horizontal equations of motion (for simplicity we have dropped the horizontal indices $i, e$ and have not included the variable porosity used with the penalization). Bottom drag and wind stress are implemented as surface fluxes in a backwards Euler split step as part of vertical diffusion (see Section 3.2). Mass flux through the air-sea

interface has been neglected, although it could be included as an extra source term in the top layer.

The first partial explicit Euler step for the scalars is

$$\mu_k^{**} = \mu_k^n - \Delta t \nabla \cdot F_k^n, \tag{10}$$

$$\Theta_k^{**} = \Theta_k^n - \Delta t \nabla \cdot (\theta_k^n F_k^n), \tag{11}$$

where $F_k^n = \mu_k^n v_k^n$ is the mass flux in each layer. Because we use Lagrangian vertical coordinates, the layer depths evolve

according to (10), and the two estimates of the depth $H + \eta^n$ and $\sum_{k=1}^N \mu_k^{**}/\rho_0$ do not agree exactly. To avoid instability associated with the inconsistent estimates of the free surface position layer dilation (Bleck and Smith, 1990) is used to stretch each layer slightly to match the free surface estimate $\eta^n$. Layer dilation is applied after each partial Euler step to correct the layer depths

$$\mu_k^* = \frac{\rho_0 (H + \eta^n)}{\sum_{k=1}^N \mu_k^{**}} \mu_k^{**}. \tag{12}$$

After dilating the layers, the mass-weighted buoyancy $\Theta_k^{**}$ is corrected using the new mass density,

$$\Theta_k^* = \frac{\mu_k^*}{\mu_k^{**}} \Theta_k^{**}. \tag{13}$$

Due to differences between the barotropic and baroclinic mass fluxes, layer dilation conserves global mass but not mass in individual layers. Nevertheless, as Hallberg and Adcroft (2009) pointed out, operational ocean models such as MICOM and

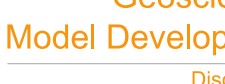 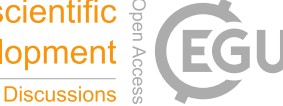

HYCOM have used this approach successfully. In any case, remapping of vertical layers also mixes buoyancy and inertial mass
between layers.

The implicit scheme for the vertical layer velocities and the free surface perturbation equation $\partial_t \eta + \nabla \cdot ((H + \eta)v) = 0$ is

$$
\begin{aligned}
v_k^{n+1} &= v_k^n + \Delta t (G_k^n - g\nabla\eta^{n+1}), & (14) \\
\eta^{n+1} &= \eta^n - \Delta t \nabla \cdot F^{n+1}, & (15)
\end{aligned}
$$

where $G_k^n$ is the right hand side of the velocity equation without the external pressure gradient and $F^{n+1} = \frac{1}{\rho_0}\sum_{k=1}^{N}\mu_k^{n+1}v_k^{n+1} :=$
$(H + \eta^{n+1})v^{n+1}$ is the depth-integrated horizontal thickness flux.

Equation (14) is first split into explicit Euler and backwards Euler steps,

$$
\begin{aligned}
v_k^* &= v_k^n + \Delta t\, G_v^n, & (16) \\
v_k^{n+1} &= v_k^* - \Delta t\, g\nabla\eta^{n+1}. & (17)
\end{aligned}
$$

We now use (17) to approximate the depth-integrated horizontal thickness flux as

$$
F^{n+1} \approx F^* - \Delta t\, g(H + \eta^n)\nabla\eta^{n+1},
$$

in (15), where $F^* = \sum_{k=1}^{N}\mu_k^* v_k^*/\rho_0$. The flux $F^{n+1}$ has been linearized about the previous value of the free surface, i.e.
$\mu_k^{n+1} \approx \mu_k^*$ and $(H + \eta^{n+1})\nabla\eta^{n+1} \approx (H + \eta^n)\nabla\eta^{n+1}$. This gives the linear elliptic equation

$$
\eta^{n+1} = \eta^n - \Delta t \nabla \cdot F^* + \Delta t^2 \nabla \cdot [g(H + \eta^n)\nabla\eta^{n+1}]. \tag{18}
$$

Rearranging and dividing by $\Delta t^2$ gives,

$$
\nabla \cdot [g(H + \eta^n)\nabla\eta^{n+1}] - \frac{\eta^{n+1}}{\Delta t^2} = -\frac{\eta^*}{\Delta t^2}, \tag{19}
$$

where we have defined the intermediate free surface

$$ \eta^* = \eta^n - \Delta t\, \nabla \cdot F^*. \tag{20} $$

The adaptive multiscale elliptic solver used to solve (19) for $\eta^{n+1}$ is described below in Section 3.3. Finally, the intermediate
layer velocities $v_k^*$ are corrected using the backwards Euler step (17) to obtain $v_k^{n+1}$.

The layer dilation correction is applied once more to $\mu_k^*$ and $\Theta_k^*$, using the new free surface perturbation $\eta^{n+1}$, to obtain
$\mu_k^{n+1}$ and $\Theta_k^{n+1}$. Note that, in contrast to the the split-explicit method, a single (slow) barotropic time step $\Delta t \approx 35\sqrt{gH}$ is
used for both the implicit and the explicit steps.

In practice, the explicit Euler steps are incorporated into an explicit RK3 or RK4 scheme (as used in the non-split time
integration option). WAVETRISK-OCEAN uses either a third or fourth-order low storage Runge–Kutta scheme (Kinnmark and
Gray, 1984). The RK3 scheme for $y' = f(y)$ is

$$
\begin{aligned}
y^1 &= y^n + \frac{\Delta t}{3} f(y^n), \\
\quad y^2 &= y^n + \frac{\Delta t}{2} f(y^1), & (21) \\
y^{n+1} &= y^n + \Delta t f(y^2).
\end{aligned}
$$

This method is third-order accurate for linear terms, second-order accurate for nonlinear terms and is stable for a CFL number less than $\sqrt{3}$. It is well-suited for large, adaptive problems because it uses only one previous time step and has low memory requirements. In a multi-step method like Runge–Kutta scheme, after each substep the layer dilation correction is applied to the intermediate values of $\mu_k$ and $\Theta_k$ and the result is interpolated back onto the adapted grid (to ensure mass conservation). The external pressure gradient is neglected in the substeps (it is included in the backwards Euler step 17, which uses the new free surface value $\eta^{n+1}$). We have checked that this time scheme preserves constants (e.g. that in the absence of remapping a constant vorticity or buoyancy field remains constant).

We finish by presenting the linear stability of the $\theta$-method, following the approach of Walters et al. (2009). This analysis specifically addresses the Coriolis term, and neglects bottom drag. Figure 2 compares the stable and unstable regions of the $\theta$ method in the $\theta - kc\Delta t$ plane for several time integration schemes, where $k$ is the perturbation wavenumber, $c = \sqrt{gH}$ is the external wave speed and $\Delta t$ is the time step. The explicit Euler and AB2 methods are both unstable for all $\theta$ at small wavenumbers, as is RK2 (not shown). In contrast, RK3 and RK4 are both stable for all $\theta \geq 0.75$. (Note that AB3 is stable for all $\theta > 1/2$ and is the current preferred choice in MITgcm.) RK3 is actually more stable than RK4 at small $k$, although this is likely not significant in practice. The results presented below use RK4 with $\theta = 1$ (i.e. fully implicit) in Sections (4.1, 4.2) and RK3 with $\theta = 0.8$ in Section 4.3.

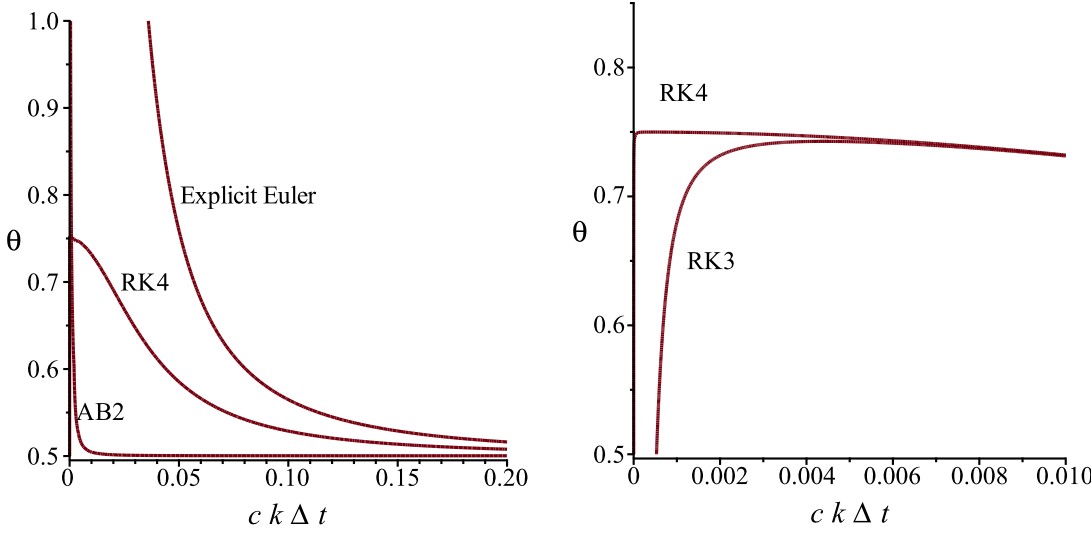

**Figure 2.** Neutral linear stability level curves in the $\theta - ck\Delta t$ plane for the $\theta$-method for several explicit schemes. The area above the red curves indicates the stable region for each scheme. Note that RK3 and RK4 are unconditionally stable for all $\theta \geq 0.75$, while AB2 and Explicit Euler are both unstable for small wavenumbers $k$.

An indication of the maximum computational efficiency of the code is given by the performance of the non-adaptive version. We have performed computations for horizontal grids $J = 7$ (163 840 cells) and $J = 8$ (655 360 cells) with 60 vertical layers for the turbulent baroclinic jet case in Section 4.3 without nudging, remapping or diffusion. We show the performance for





different choices of patch size $p$ for the hybrid data structure. (Patches are the lowest level of the quad tree, and are uniform $2^p \times 2^p$ grids.) All runs were performed on the Compute Canada machine `niagara` with 40-core Intel Skylake nodes, where each node has 202 GB of memory.

Table 1 summarizes the metric $\tau$ = (wall clock time × cores) / (iterations × nodes × vertical layers), where iterations = 3 for RK3. For the explicit scheme the best performance is $\tau \approx 0.8\,\mu s$, while for the split time scheme the best performance
is $\tau \approx 1\,\mu s$. As a comparison with the mode-split case, the best performance of the highly optimized regional ocean model ROMS (Shchepetkin and McWilliams, 2005) is a bit larger than 1 μs (Roullet, 2019) for realistic configurations, or slightly less than 1 μs with only the dynamical core (as here). (Note that a global model like WAVETRISK-OCEAN has some additional over- head associated with the spherical topology.) Thus, WAVETRISK-OCEAN has a similar computational performance to ROMS when run non-adaptively.

| Grid level $J$ | patch size | 20 cores | | 40 cores | | 160 cores | |
|---|---|---|---|---|---|---|---|
| | | explicit | split | explicit | split | explicit | split |
| 7 (163 840 cells) | $8 \times 8$ | 1.23 μs | 1.33 μs | 1.42 μs | 1.46 μs | 1.30 μs | 1.26 μs |
| 7 (163 840 cells) | $16 \times 16$ | 0.816 μs | 1.05 μs | 0.850 μs | 1.08 μs | 0.993 μs | 1.29 μs |
| 8 (655 360 cells) | $8 \times 8$ | 1.74 μs | 1.86 μs | 1.95 μs | 2.13 μs | 1.93 μs | 2.03 μs |
| 8 (655 360 cells) | $32 \times 32$ | 0.771 μs | 0.961 μs | 0.708 μs | 0.964 μs | 0.869 μs | 1.09 μs |

**Table 1.** Computational performance of the explicit and barotropic-baroclinic mode split time schemes without nudging, remapping or diffusion for non-adaptive runs with 60 vertical layers for a modified version of the turbulent baroclinic jet case discussed in Section 4.3. Patch size is the size of the uniform patches in the hybrid data structure (i.e. the lowest level of the quad tree). The metric used is (wall clock time × cores) / (iterations × nodes × vertical layers), where iterations = 3 for RK3.

For the 60 vertical layer case considered here, the mode split scheme adds an overhead of 3–30%. Since it uses time steps about 45 times larger, the mode split version of the code is about 34–44 times faster than the explicit scheme. The overhead associated with adaptivity depends on the number of refinement levels, load balancing, how often the grid is adapted, the selected tolerance, and the patch size. For a well-balanced case with a grid compression of about 10 times, adaptive runs are about 1.5 times slower *per active node* than non-adaptive runs on a single grid level (Kevlahan and Dubos, 2019, confirmed for
the mode split case). In practice, the performance of realistic, well-balanced, adaptive runs with at least O($10^6$) active nodes is about $\tau$ = O(1 μs).

## 3.2 Vertical diffusion and TKE closure

WAVETRISK-OCEAN implements Laplacian vertical diffusion of buoyancy (i.e. the thermodynamic variable) and velocity in each vertical column as a backwards Euler split step after the main time step. This implicit method is unconditionally stable.
The diffusion coefficients of buoyancy and velocity, $K_t$ and $K_m$, are evaluated either analytically (see the upwelling test case 4.2) or using an eddy viscosity model with a Kolmogorov-type closure of the TKE. The TKE closure is similar to that





used in the NEMO ocean model (Madec and Team, 2015, section 10.1.3). TKE is computed dynamically in each vertical column using the one-dimensional equation

$$\partial_t e_{il} = K_m \|\partial_z v_{ek}\|^2 - K_t N_{il}^2 + \partial_z (K_m \partial_z e_{il}) - c_\epsilon \frac{e_{il}^{3/2}}{l_\epsilon}, \tag{22}$$

where the TKE $e_{il}$ is defined at node $i$ and interface $0 \leq l \leq N$, $N_{il}^2 = -g\delta_l[\rho_{ik}]/\rho_0$ is the local Brunt–Väisälä frequency squared and $l_\epsilon$ is the dissipation length scale. $\|\partial_z v_{ek}\|^2$ is computed at nodes using the usual WAVETRISK formula for kinetic energy applied to $\partial_z v_e$. The eddy viscosity $K_m$ and eddy diffusivity $K_t$ are then found from the TKE (dropping indices) as

$$K_m = \max(c_m l_m \sqrt{e}, K_{m0}), \qquad K_t = \max(K_m/\text{Pr}_t, K_{t0}), \tag{23}$$

where $c_m = 0.1$, $l_m$ is the mixing length and $K_{m0}, K_{t0}$ are minimum diffusivities. The Prandtl and Richardson numbers are

$$\text{Pr}_t = \begin{cases} 1 & \text{if Ri} < 0.2, \\ 5\text{Ri} & \text{if } 0.2 \leq \text{Ri} \leq 2, \\ 10 & \text{if Ri} > 2, \end{cases} \qquad \text{Ri} = \frac{N^2}{\|\partial_z v\|^2 + \varepsilon_s}, \tag{24}$$

where $\varepsilon_s = 10^{-20}\,\text{s}^{-2}$. The length scales are computed as in NEMO from intermediate values $l_{\text{up}}$ and $l_{\text{dwn}}$ to ensure that their maximum vertical gradients are not larger than depth variations. This modifies the initial values from the basic formula

$$l_m = l_\epsilon = \sqrt{\frac{e}{\max(N^2, N_\varepsilon^2)}},$$

where $N_\varepsilon^2 = 10^{-20}\,\text{s}^{-2}$. The Dirichlet boundary conditions for TKE are

$$e(z = \eta, t) = \max(C_{\text{sfc}} \|\tau\|/\rho_0, e_0^{\text{sfc}}), \qquad e(z = -H, t) = e_0,$$

where $C_{\text{sfc}} = 67.83$, $\tau$ is the surface wind stress, $e_0^{\text{sfc}} = 10^{-4}\,\text{m}^2/\text{s}^2$ and $e_0 = 10^{-6}/\sqrt{2}\,\text{m}^2/\text{s}^2$. This large value of $C_{\text{sfc}}$ (compared with the usual value of 3.75), together with a modification of the length scale computation, parameterizes the effect of surface wave breaking.

The TKE equation (22) is advanced in time from $n$ to $n+1$ using an implicit backwards Euler step, discretized as,

$$e_{il}^{n+1} = e_{il}^n + \Delta t \left\{ K_m \|\partial_z v_{ek}^n\|^2 - K_t N_{il}^2 \right\} + \Delta t \left\{ \frac{1}{\Delta z_{il}} \delta_l \left[ \frac{K_m}{\Delta z_{ik}} \delta_k \left[ e_{il}^{n+1} \right] \right] - c_\epsilon \frac{\sqrt{e_{il}^n} e_{il}^{n+1}}{l_\epsilon} \right\}. \tag{25}$$

Positivity of TKE is guaranteed by discretizing the buoyancy term implicitly by multiplying it by $e_{il}^{n+1}/e_{il}^n$ when the source term (the second term on the right hand side) is negative, i.e. the "Patankar trick" (Patankar, 1980). (Note that $N^2$ is always evaluated at time step $n$.) The resulting one-dimensional tridiagonal system is solved using the lapack routine dgtsv.

After the eddy viscosity and eddy diffusivity have been updated, vertical diffusion is applied to the buoyancy and velocity
using a backwards Euler split step,

$$\theta_{ik}^{n+1} = \theta_{ik}^n + \frac{\Delta t}{\Delta z_{ik}} \delta_k \left[ \frac{K_t}{\Delta z_{il}} \delta_l [\theta_{ik}^{n+1}] \right], \tag{26}$$

$$v_{ek}^{n+1} = v_{ek}^n + \frac{\Delta t}{\Delta z_{ek}} \delta_k \left[ \frac{K_m}{\Delta z_{el}} \delta_l [v_{ek}^{n+1}] \right]. \tag{27}$$





Source terms at the free surface and bottom (e.g. wind stress, bottom friction, heating/cooling) are implemented via the appropriate Neumann (i.e. vertical flux) boundary conditions. Note that surface heat flux boundary conditions for the temperature,

$F_T = Q/(\rho_0 c_p)$, becomes $F_\theta = Q/(\rho_0 c_p)a_0/\rho_0$ using the simple linear equation of state (2) (without salinity or representation of thermobaric and cabbeling effects).

The numerical implementation of vertical diffusion (26,27) and the associated TKE closure scheme (25) has been verified using two standard one-dimensional test cases: boundary layer thickening (Kato and Phillips, 1969) and free convection (Willis and Deardorff, 1974). In these cases only the vertical diffusion is active, and the code is run at a coarse resolution $J = 4$. In

both cases the results matched exactly those produced by NEMO using the same TKE closure model.

The current version of WAVETRISK-OCEAN also includes an enhanced buoyancy diffusion option and a solar penetrative flux model, as in NEMO (Madec and Team, 2015, section 5.4.2). The NEMO model is based on a two-waveband light penetration scheme.

### 3.3 Adaptive multiscale elliptic solver

The barotropic–baroclinic mode splitting relies on an efficient and sufficiently accurate algorithm for solving the associated two-dimensional elliptic problem (19). The implicit free surface method is computationally efficient since, unlike the rigid lid method, it does not require a very accurate solution for the free surface perturbation $\eta$ to achieve an accurate representation of the slow baroclinic and vortical modes. The WAVETRISK algorithm provides a natural adaptive multiscale set of approximation subspaces that we can take advantage of in a simple multigrid elliptic solver (Vasilyev and Kevlahan, 2005).

The elliptic equation is first solved to high accuracy on the coarsest grid $J_{min}$ using bicgstab. A relative residual norm error of $10^{-9}$ (Kang et al., 2021) is achieved in 20–30 iterations with a barotropic CFL condition of 35 (or in 5–10 iterations with a CFL condition of 10). The solution is then prolonged to the next finer level $J_{min} + 1$ using the the standard WAVETRISK interpolation operator for scalars, and the solution is improved using 20–60 Jacobi iterations (a larger residual tolerance is sufficient at these finer scales). This process is continued until the solution is obtained on the finest grid. Since there are relative few active grid

points on the finer grids, this simple multiscale elliptic solver is quite fast.

To accelerate the Jacobi iterations we take advantage of the scheduled relaxation Jacobi (SRJ) method (Yang and Mittal, 2014). We use 30 distinct optimal relaxation factors computed for the elliptic equation (19) using the Chebyshev–Jacobi variant of SJR (Adsuara et al., 2017). This method reduces the residual error at the finest scales by six orders of magnitude about eight times faster than the standard Jacobi method, with no additional overhead.

### 325 3.4 Penalization of lateral boundaries

Kevlahan et al. (2015) introduced a volume penalization to approximate complex multiscale topography for the two-dimensional shallow water equations. This method uses variable porosity $\phi(\boldsymbol{x})$ and permeability $\sigma(\boldsymbol{x})$ to approximate no-slip boundary conditions in the limit $\phi \to 0$ and $\sigma \to 0$. Solid regions are defined using a mask function $\chi(\boldsymbol{x})$, which equals 1 in solid regions and equals 0 in fluid regions. In practice, the mask is smoothed over a few grid points.



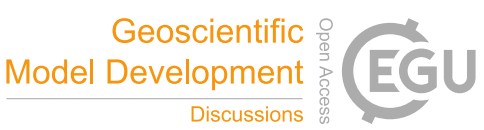

Since penalization defines solid regions implicitly by modifying the equations, it is especially well-suited for complicated geometries in dynamically adaptive methods since the coastal geometry can be refined easily as the local grid resolution changes. This avoids having to restrict the maximum resolution of the geometry or, conversely, carry extremely fine grids along the coast even when not justified by the fluid dynamics. Kevlahan et al. (2015) showed that the error in satisfying the boundary condition is $O(\alpha\epsilon^{1/2})$, where $\alpha$ and $\epsilon$ are, respectively, the porosity and permeability in the solid regions. Guinot and

Soares-Frazao (2006); Guinot et al. (2018) have developed a similar penalization method for modelling coastal inundation in urban environments (including subgrid scale modelling of unresolved topography).

Debreu et al. (2020) developed a three-dimensional extension of this volume penalization to represent bottom bathymetry and non-vertical lateral boundaries. However, in the present paper we restrict ourselves to vertical lateral boundaries and represent bathymetry via a hybrid grid that is approximately uniform in $z$ in shallow regions and terrain following in deep

regions Shchepetkin and McWilliams (2009). We intend to implement the fully three-dimensional penalization in future work, and concentrate here on developing and validating a basic dynamically adaptive barotropic–baroclinic mode splitting global ocean model.

In the results presented here we fix the porosity in the solid $\alpha = 0.01$ and the permeability $\epsilon = \Delta t$ (minimum stable value for an explicit time step). The velocity penalization is applied in a split step, after the main time step, as in Rasmussen et al.

345    (2011).

### 3.5  Summary of the complete algorithm

We complete the presentation of the WAVETRISK-OCEAN algorithm by briefly summarizing its main steps in Algorithms 1–4.

---

**Algorithm 1** Complete WAVETRISK-OCEAN time stepping algorithm.

---

$t = 0$

**while** $t \leq T$ **do**

   **Set time step** $\Delta t$ {use strictest of barotropic, baroclinic and advective CFL conditions}

   **Explicit Runge–Kutta step** (Algorithm 2)

   **Implicit free surface step** (Algorithm 3)

   **Vertical diffusion split step** (Section 3.2)

   **Conservative remapping** (Engwirda and Kelley, 2016) {every 5–20 time steps}

   **Wavelet transform cycle** (Algorithm 4)

   **Adapt horizontal grid** (Section 2.2)

   **Apply boundary condition penalization** (Section 3.4)

   $t = t + \Delta t$

**end while**

---





---

**Algorithm 2** Explicit Runge–Kutta sub-cycles (see Equation 21). The steps below are repeated three times for RK3 and four times for RK4.

---

    **Forward Euler step** (Equations 10,11,21)

    **Layer dilation** corrections (Equations 12, 13)

    **Wavelet transform cycle** (Algorithm 4)

---

**Algorithm 3** Implicit free surface correction.

---

    **Solve elliptic equation** for new free surface (Equation 19, Section 3.3)

    **Backwards Euler step** to correct velocity (Equation 17)

    **Layer dilation** corrections (Equations 12, 13)

    **Wavelet transform cycle** (Algorithm 4)

---

## 4 Results

### 4.1 Seamount test case

The seamount test case was introduced by Beckmann and Haidvogel (1993) to quantify the pressure gradient errors in a $\sigma$ vertical coordinate system where the vertical layers are stretched between the sea floor and the free surface. This test case consists of a tall Gaussian bathymetry profile with a flat density perturbation that decreases exponentially with depth. In $\sigma$ coordinates the vertical layers are therefore not aligned with the horizontal isopycnals. The axisymmetric bathymetry is defined as

$$h(r) = H(1 - De^{-r^2/L^2}),\tag{28}$$

with $H = 5\,\mathrm{km}$, $D = 0.9$, $L = 40\,\mathrm{km}$. The initial density profile with stable horizontal stratification is

$$\rho(z) = \rho_0 + \delta\rho\, e^{z/\delta},\tag{29}$$

$\delta = 500\,\mathrm{m}$ and $\rho_0 = 1000\,\mathrm{kg/m^3}$. For this configuration the Brunt–Väisälä frequency is defined as

$$N_0^2 = -\frac{g\,\delta\rho}{\rho_0 H_0},$$

and the Burgers number

$$S = \frac{N_0 H_0}{f_0 L}.$$

---

**Algorithm 4** Wavelet transform cycle.

---

    **Compute wavelets** of all variables

    **Zero out wavelets** less than threshold $\varepsilon$

    **Inverse wavelet transform** onto adapted grid {conserves mass}

---





The original test case was formulated for an $f$-plane approximation. We have extended this test case to the sphere by placing the centre of Gaussian seamount at latitude 43.29 N such that $f = 10^{-4}\,\mathrm{s}^{-1}$. The radius of the planet is $a \approx 153\,\mathrm{km}$, its rotation rate $\Omega = 7.2921 \times 10^{-5}\,\mathrm{rad/s}$ and the kinematic viscosity is $\nu = 50\,\mathrm{m}^2/\mathrm{s}$.

We compare the growth of the spurious velocity for three different stratifications with Burgers numbers $S = 0.5$, 1.5 and 3 (corresponding to $\delta\rho = -0.0816\,\mathrm{kg/m}^3$, $-0.735\,\mathrm{kg/m}^3$ and $-3\,\mathrm{kg/m}^3$). For all cases we use 20 vertical layers and an nonadaptive horizontal grid with fixed resolution level $J_{\min} = 5$ ($\Delta x = 5.75\,\mathrm{km}$) to set the maximum topographic stiffness ratio

$$r_{\max} = \frac{|h_{i+1} - h_i|}{h_i + h_{i+1}} \approx 0.21.$$

This value is close to the maximum value typically allowed in operational models to ensure acceptable pressure gradient error. The vertical grid uses Chebyshev nodes, which concentrate the vertical layers at the free surface and sea floor. A uniform vertical grid gives similar results, but with a slightly higher pressure gradient error. The vertical grid is remapped to the original Chebyshev nodes every time step. A constant longitude slice through the computational grid is shown in figure 3 (top) and the corresponding initial stratification is shown in figure 3 (bottom). The barotropic CFL number is fixed at $C_{\mathrm{barotropic}} = 15$ for all simulations, corresponding to $\Delta t = 347\,\mathrm{s}$. The baroclinic CFL numbers for the three stratifications are therefore $C_{\mathrm{baroclinic}} = c_1 \Delta t / \Delta x = 0.04$, 0.13 and 0.26. All simulations are run for 40 days, significantly longer than the 10 day results reported in Beckmann and Haidvogel (1993).

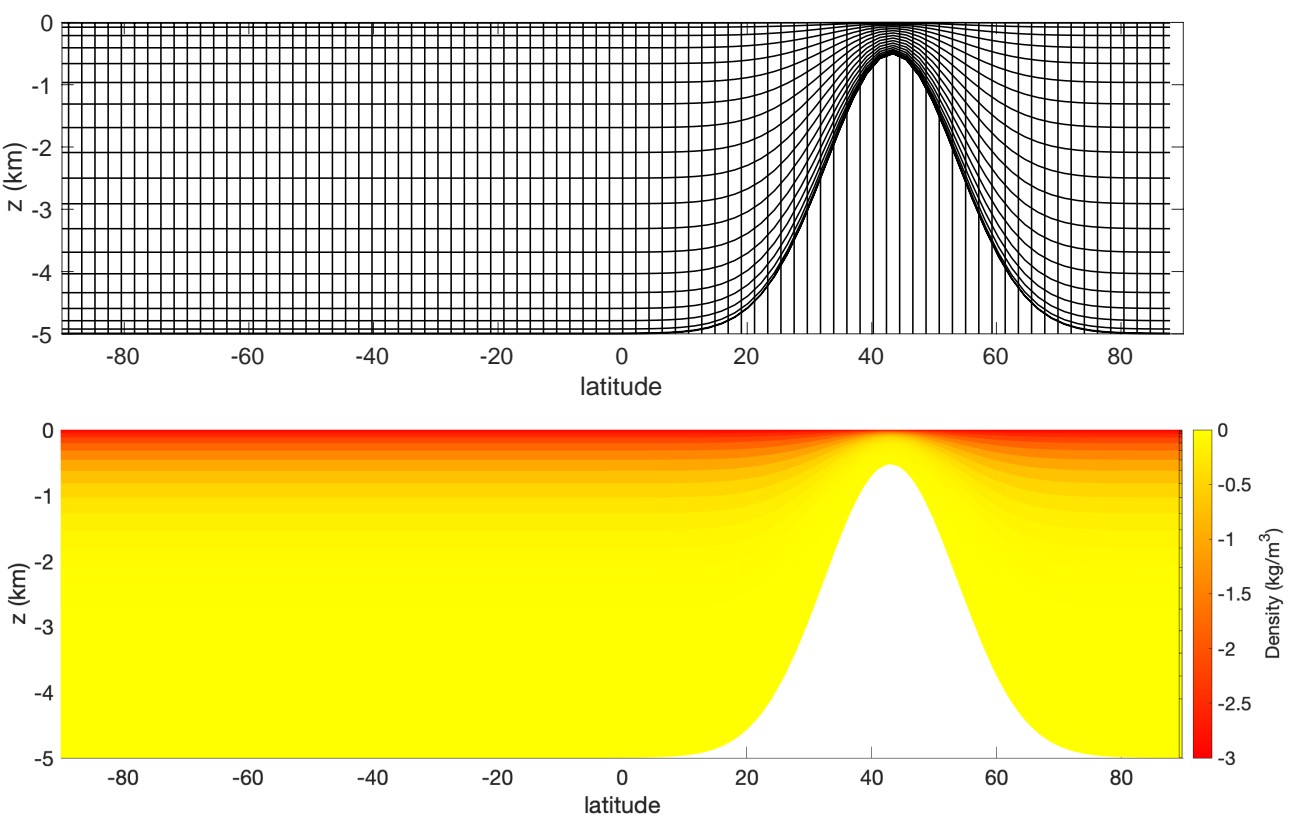

**Figure 3.** $\sigma$-Chebyshev grid (top) and initial stratification (bottom) for the seamount test case with $\delta\rho = -3 \ \mathrm{kg/m^3}$.

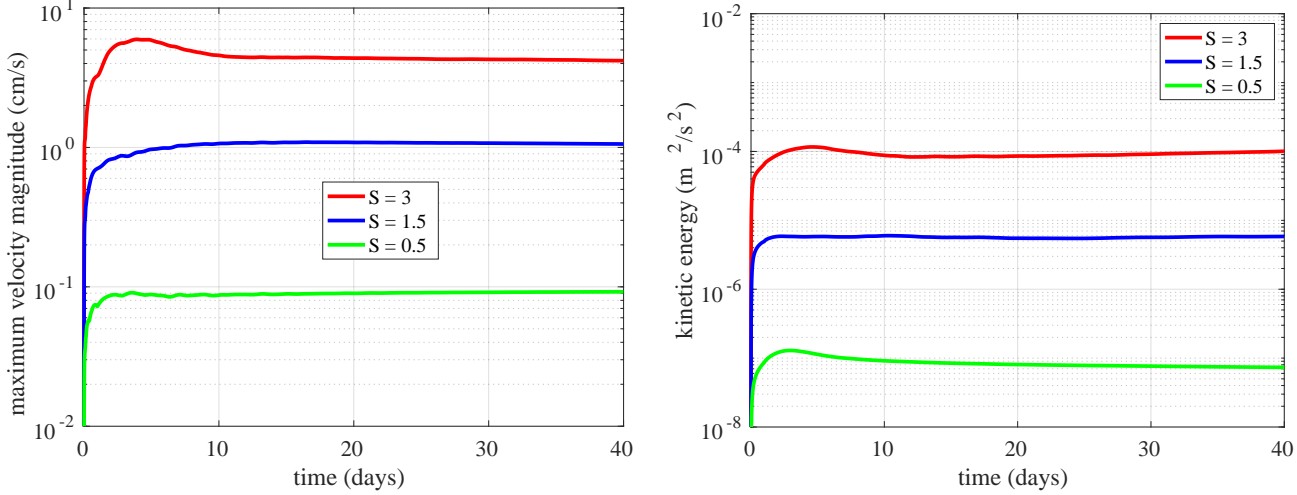

**Figure 4.** Maximum velocity magnitudes (left) and kinetic energies (right) for the seamount test case for three Burgers numbers. The maximum topographic stiffness ratio $r_{\max} = 0.21$



| $a$ | $240\,\text{km}$ | $L$ | $80\,\text{km}$ |
|---|---|---|---|
| $\rho_0$ | $1027\,\text{kg/m}^3$ | $g$ | $9.806\,16\,\text{m/s}^2$ |
| $f_0$ at $45°$ | $-8.4853 \times 10^{-5}\,\text{rad/s}$ | | |
| $H$ | $150\,\text{m}$ | $H_{\min}$ | $25\,\text{m}$ |
| $r_d$ | $3 \times 10^{-4}\,\text{m/s}$ | $\tau_0$ | $-0.1\,\text{m/s}$ |

**Table 2.** Parameters for the upwelling test case: reference density $\rho_0$, Coriolis parameter $f_0$, gravitational acceleration $g$, wind stress $\tau_0$, bottom friction $r_d$, planetary radius $a$, minimum depth $H_{\min}$, maximum depth $H$, channel meridional width $L$.

Figure 4 shows that the maximum spurious velocities quickly stabilize at $0.092\,\text{cm/s}$, $1.1\,\text{cm/s}$ and $4.3\,\text{cm/s}$ for Burgers numbers $S = 0.5$, $1.5$ and $3.0$ respectively. If $r_{\max} = 0.38$, but the time step is kept fixed at $\delta t = 347\,\text{s}$, the spurious velocity magnitude increases only slightly to $5.11\,\text{cm/s}$ for the $S = 3$ case. In addition to verifying the barotropic–baroclinic splitting algorithm and the incompressible version of the DYNAMICO discretization with the non-adaptive runs, we also confirmed that allowing three levels of grid refinement does not amplify the spurious velocity fields.

These results are similar to Debreu et al. (2020), who find a maximum velocity magnitude of about $7.5\,\text{cm/s}$ for $S = 3$ with $r_{\max} = 0.21$ using the regional CROCO model with $\Delta x = 3\,\text{km}$. Beckmann and Haidvogel (1993) reported maximum velocities of $0.987\,\text{cm/s}$ ($S = 0.5$), $1.255\,\text{cm/s}$ ($S = 1.5$) and $1.329\,\text{cm/s}$ ($S = 3.0$) after 10 days for $r_{\max} = 0.21$ with a stretched horizontal grid.

Shchepetkin and McWilliams (2003) found larger spurious velocities, however they chose a larger maximum topographic stiffness ratio ($0.29$ compared to our value of $0.21$) and used only 10 vertical layers and $\Delta x = 6.7\,\text{km}$. Their best result for $S = 3.1$ was a maximum velocity magnitude of about $10.8\,\text{cm/s}$ at 10 days, decreasing to about $7\,\text{cm/s}$ by 120 days.

These results confirm that the $n$-$\text{IL}^0$ provides good control of the pressure gradient error.

## 4.2 Upwelling test case

This is based on the standard ROMS test case contributed by Macks and Middleton. It models wind-driven coastal upwelling/downwelling in a periodic channel with stable stratification. We have adapted the test case to the sphere by considering a zonal channel of width $80\,\text{km}$ centred at latitude $\phi_0 = 45°$ and maximum depth $H = 150\,\text{m}$ on a small planet of radius $240\,\text{km}$ (see figure 5 (right)). The land mass is implemented using volume penalization with porosity $\alpha = 10^{-6}$. The parameters for this test case are summarized in table 2.

The profile of the zonal channel is given in terms of latitude $\phi$ by

$$z(\phi) = \begin{cases} -H + A\left[1 - \tanh\left(\frac{1}{\Delta}(f(y) - \frac{L}{8})\right)\right], & |\phi - \phi_0| \leq \frac{\delta\phi}{2} \\ -H_{\min}, & \text{otherwise} \end{cases}$$

with

$$f(y) = \begin{cases} y & y \leq L/2, \\ L - y & \text{otherwise}, \end{cases}$$



| Layer | $z$ (m) | $\Delta z$ (m) | $\rho(z)$ (kg/m$^3$) |
|---|---|---|---|
| 1 | -127 | 45.9 | 1 0282.2 |
| 2 | -89.1 | 30.2 | 1 0281.5 |
| 3 | -6.40 | 2.00 | 1 0281.0 |
| 4 | -4.72 | 1.35 | 1 0278.9 |
| 5 | -35.8 | 9.26 | 1 0270.2 |
| 6 | -27.9 | 6.53 | 1 0261.0 |
| 7 | -22.3 | 4.77 | 1 0258.4 |
| 8 | -18.1 | 3.64 | 1 0257.9 |
| 9 | -14.8 | 2.90 | 1 0257.7 |
| 10 | -12.2 | 2.43 | 1 0257.6 |
| 11 | -9.88 | 2.12 | 1 0257.6 |
| 12 | -7.85 | 1.93 | 1 0257.6 |
| 13 | -5.98 | 1.81 | 1 0257.5 |
| 14 | -4.22 | 1.73 | 1 0257.5 |
| 15 | -2.51 | 1.69 | 1 0257.4 |
| 16 | -0.833 | 1.67 | 1 0257.4 |

**Table 3.** Vertical layers at the centre of the zonal channel: layer centre $z$, layer thickness $\Delta z$ and density $\rho(z)$.

with $A = (H - H_{\min})/(1 - \tanh\left(-\frac{L}{8\Delta}\right))$, $\Delta = 57\,\mathrm{km}$, $y = \frac{\pi a}{180}\left(\phi - (\phi_0 - \delta\phi)\right)$, channel meridional width $\delta\phi = \frac{L}{a}\frac{180}{\pi} \approx 19°$. The channel profile is shown in figures 5 (left) and 6.

The vertical grid is hybrid $z - \sigma$ grid that approximates a uniform in $z$ grid in shallow regions, and at $\sigma$ grid in deep regions (see 5 left). This grid is similar to the hybrid grid described in Shchepetkin and McWilliams (2009), and available in NEMO.

The stably stratified temperature profile is given by

$$T(z) = T_a + 4\tanh\left(\frac{z - z_0}{h_z}\right) + \frac{z - z_1}{\tilde{H}},$$

with $T_a = 14\,°\mathrm{C}$, $h_z = 6.5\,\mathrm{m}$, $z_0 = -35\,\mathrm{m}$, $z_1 = -75\,\mathrm{m}$, $\tilde{H} = 150\,\mathrm{m}/°\mathrm{C}$. Density (and therefore buoyancy) depends on temperature via a linear equation of state

$$\rho(z) = \rho_0 - a_0(T(z) - T_a),$$

with $a_0 = 0.28\,\mathrm{kg/m}^3/°\mathrm{C}$. The vertical layers and densities in the centre of the channel are given in table 3.

Three different simulations were computed: a non-adaptive simulation with resolution $J = 8$ ($\overline{\Delta x} = 1\,\mathrm{km}$ (comparable to the resolution $\Delta x = 1.25\,\mathrm{km}$ of the CROCO benchmark simulation), and two adaptive simulations with resolutions $J = 6, 7, 8$

(low resolution) and $J = 8, 9, 10$ (high resolution). Note that for the low resolution simulation the topographic stiffness ratio $r_{\max} = 0.66$ at the coarsest resolution, which is much larger than the value $r_{\max} < 0.2$ to ensure acceptable pressure gradient error. (A resolution of at least $J = 8$ is required to achieve an acceptable $r_{\max} = 0.17$.) Thus, the low resolution run verifies the





ability of the adaptive code to use much coarser grids than is possible for a non-adaptive code. The high resolution run tests the

ability of the code to provide local high resolution ($0.25\,\mathrm{km}$) where needed to achieve more accurate results. The CFL number

$C_{\mathrm{barotropic}} = 35$, which corresponds to a time step $\Delta t = 918\,\mathrm{s}$ at resolution $J = 8$. The vertical grid is remapped to the initial

$z - \sigma$ grid every $10\Delta t$.

Laplacian vertical diffusion of momentum and temperature is implemented via a backwards Euler implicit split step. The

eddy diffusion $K_t = 10^{-6}\,\mathrm{m^2/s}$ is constant and the depth-dependent $K_v(\mathbf{x}, z)$ is give by

$$K_v(\mathbf{x}, z, t) = K_0 \left( 1 + 4 \exp\left( \frac{z - \eta(\mathbf{x}, t)}{H} \right) \right),$$

where $K_0 = 2 \times 10^{-3}\,\mathrm{m^2/s}$ and $\eta(\mathbf{x}, t)$ is the free surface perturbation.

The results are shown in figure 6, compared with the benchmark CROCO simulation on the $f$-plane with $f_0 = -8.26 \times 10^{-5}\,\mathrm{s^{-1}}$.

Note that since WAVETRISK-OCEAN uses a Lagrangian vertical grid, we first remap to the initial grid, diagnose vertical velocity

from the volume flux $\Omega$ through the interfaces, and finally add the component of the pseudo-horizontal velocity in the vertical

direction to obtain the true vertical velocity. All results are zonal averages.

The WAVETRISK-OCEAN results are qualitatively similar to the CROCO results, although the maximum zonal velocity is

higher (about $34\,\mathrm{cm/s}$, very similar to the ROMS upwelling test case result, https://www.myroms.org/wiki/UPWELLING_

CASE). Comparing the non-adaptive J8 results to the adaptive J6J8 results shows that the adaptive code is able to reproduce

the main quantitative and qualitative features of a non-adaptive simulation at the highest resolution. This shows that dynamic

adaptivity can overcome limitation imposed by the topographic stiffness ratio, $r_{\mathrm{max}} < 0.2$, by using higher resolutions only

where the bathymetry gradients are large (see figure 5 right). The main difference is that the maximum zonal velocity at low

latitudes (lower Coriolis) extends to greater depths.

These results confirm that our code is able to correctly reproduce the physics of coastal upwelling in an idealized configu-

ration, taking into account the differences between simulations on the sphere and on an $f$-plane (variable Coriolis force, much

longer zonal channel width, no-slip boundary conditions).

In our final test case, we consider a more realistic baroclinic jet configuration with a more sophistical eddy viscosity model

for vertical diffusion, based on a turbulent kinetic energy closure, similar to that used in NEMO.



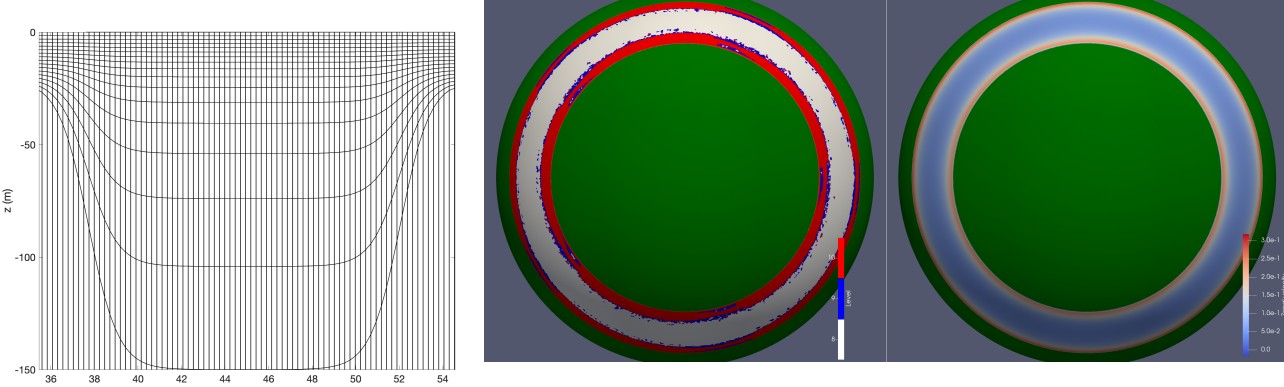

**Figure 5.** Upwelling test case. Left: vertical grid, with mean horizontal spacing at resolution $J = 8$. Right: horizontal adaptive grid and zonal velocity at day 2 at vertical level 8. The levels $j = 8, 9, 10$ correspond to mean resolutions $\overline{\Delta x} = 1\,\mathrm{km}$, $0.5\,\mathrm{km}$, $0.25\,\mathrm{km}$.



**Figure 6.** Results for the upwelling test case: zonal averages at day 2. Note that the CROCO results are for a $\beta-$plane with zonal channel width of only 20 km and the figures do not show the lowest vertical layer.

## 4.3 Baroclinic jet test case

The final test case assesses the ability of WAVETRISK-OCEAN to simulate submesoscale dynamics. The configuration is a version of the unstable baroclinic jet in a zonal channel proposed by Soufflet et al. (2016), modified for spherical geometry. This test case is designed to include the two dominant mechanisms for generating upper ocean turbulence: surface density stirring by mesoscale eddies and fine scale instabilities that drive submesoscale turbulence. The original configuration is on





a $\beta$-plane with Coriolis frequency $f_0 = 10^{-4}\,\mathrm{s}^{-1}$ and $\beta = 1.6 \times 10^{-11}\,\mathrm{m}^{-1}\mathrm{s}^{-1}$. The physical domain is a zonally periodic
channel of size $500\,\mathrm{km}$ by $2000\,\mathrm{km}$ with a uniform depth of $4000\,\mathrm{m}$ and free slip boundary conditions in the meridional
direction. The Rossby deformation radius is $\approx 30\,\mathrm{km}$. The initial density perturbation is zonally invariant, with meridional and
depth dependent gradients. The initial velocity is chosen such that it is in geostrophic balance with the density gradient (i.e.
integrating upwards, assuming a geostrophic thermal wind balance and zero velocity at the bathymetry). Soufflet et al. (2016)
consider four (fixed) grid resolutions, $\Delta x = 20\,\mathrm{km}, 10\,\mathrm{km}, 5\,\mathrm{km}, 2\,\mathrm{km}$, with vertical resolutions of 40, 60, 80 and 100 layers
respectively. Since there is no wind stress forcing, energy is maintained by nudging the zonally averaged velocity and density
to their initial profiles, with a relaxation time of 50 days.

We have adapted this baroclinic test case to the sphere by considering a small planet of radius $a = 1000\,\mathrm{km}$, with a zonal
channel of meridional width $1000\,\mathrm{km}$ centred at 30° N. No-slip boundary conditions are implemented at the channel walls,
using the penalization method described in Section 3.4. Because WAVETRISK-OCEAN is adaptive, using a relatively large grid
$J_{\min} = 5, \Delta x_{\max} \approx 38\,\mathrm{km}$, for the coarsest resolution ensures that few grid points are used in the solid (penalized) regions. We
allow four levels of grid refinement, $J = 6, 7, 8, 9$, which corresponds to a minimum resolution $\Delta x_{\min} \approx 2.1\,\mathrm{km}$. The simulation
uses 60 vertical hybrid layers, ranging in thickness from $430\,\mathrm{m}$ to $2.5\,\mathrm{m}$ at the free surface. The time step $\Delta t = 370\,\mathrm{s}$,
equivalent to CFL numbers $C_{\mathrm{barotropic}} = 35$ and the maximum $C_{\mathrm{baroclinic}} = 1.2$ (for internal waves). Since the maximum velocity
is about $75\,\mathrm{cm/s}$, the corresponding advective CFL number is about 0.14. In fact, the simulations are stable and the results are
very similar for $\Delta t \leq 630\,\mathrm{s}$.

The Lagrangian vertical grid is remapped every $5\Delta t$ to the original hybrid grid, and the horizontal grid is adapted every $\Delta t$
with a relative tolerance $\varepsilon = 0.02$ for all variables. Vertical diffusion is implemented using the TKE closure model described in
Section 3.2. Horizontal bilaplacian diffusion is included, with viscosities $\nu = 2.61 \times 10^8\,\mathrm{m}^4/\mathrm{s}$ for the densities and divergent
mode, and $\nu = 1.63 \times 10^7\,\mathrm{m}^4/\mathrm{s}$ for the rotational mode. A small amount of Laplacian diffusion, with viscosity $\nu = 5\,\mathrm{m}^2/\mathrm{s}$, is
applied to the free surface after the elliptic solve, but before the external pressure gradient correction.

The nudging is implemented by computed the current zonally averaged velocity profiles at the coarsest level $J_{\min} = 5$, and
then interpolating the required nudging to each active grid point.

The initial geostrophically balanced density and zonal velocity profiles are shown in Figure 7. The velocity magnitude is
about 3.5 times larger than in Soufflet et al. (2016) due to the more intense horizontal density gradient in the narrower channel.
The spherical geometry and longer zonal channel length also mean the results differ quantitatively from Soufflet et al. (2016).



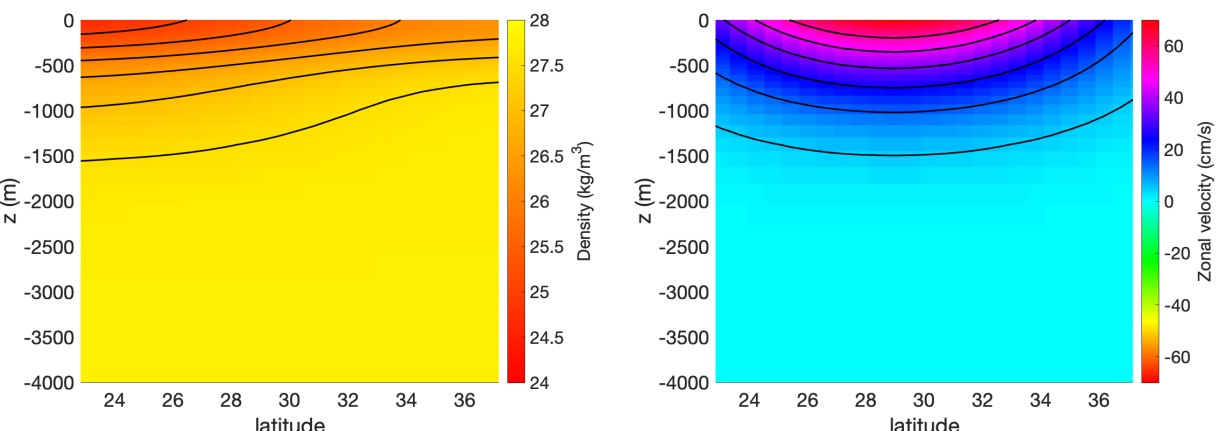

**Figure 7.** The initial geostrophically balanced density and zonal velocity profiles for the baroclinic jet case. Left: initial density anomaly $\rho - 1000\,\mathrm{kg/m^3}$. Contours at levels $25, 25.5, 26, 26.5, 27, 27.5\,\mathrm{kg/m^3}$. Right: zonal velocity. Contours at levels $5, 15, 25, 35, 45, 55\,\mathrm{cm/s}$.

WAVETRISK-OCEAN was run on 160 cores of the `compute canada` machine `niagara`. To spin up, the code was first run non-adaptively at resolution $J = 5$ for 300 days and then restarted from the checkpoint with the four additional adaptive levels and a relative tolerance $\varepsilon = 0.02$. Our goal is to have a well-developed turbulent flow to assess the adaptivity and energy spectra, not to compute climate statistics, and so the 20 year run of Soufflet et al. (2016) is not necessary. Since our domain is

12.6 times longer in the zonal direction than the domain used in Soufflet et al. (2016), the ergodic hypothesis could be used to compute statistical quantities using spatial averages instead of temporal averages (provided the flow has reached a statistically stationary state).

Figure 8 shows the adapted grid, density perturbation and relative vorticity near the surface at depth $z = -1.25\,\mathrm{m}$ at 600 days. The baroclinic jet has become unstable and generate strong submesoscale turbulence. The relatively tolerance is small, $\varepsilon =$

0.02, activating the finest resolution in areas of active turbulence. The green (land) regions use the coarsest grid. To illustrate the effect of a larger tolerance, figure 9 shows the results with $\varepsilon = 0.06$. The grid is far more compressed, with large areas requiring only the coarsest grid. Overall, the $\varepsilon = 0.06$ case uses about half as many grid points than the $\varepsilon = 0.02$ case, while still capturing the intense small scale structures in the density and vorticity fields.

For simplicity, energy spectra are computed from saved vorticity checkpoint data interpolated to fill a fine level of resolution

(e.g. $J_{\max}$ or $J_{\max} - 1$). This non-adaptive spherical data on a non-uniform hexagonal grid is then projected onto a uniform longitude–latitude grid of equivalent resolution. The spherical harmonics energy spectrum is then computed from the latitude–longitude data using the spherical harmonics toolbox SHTOOLS (Wieczorek and Meschede, 2018). In addition to global energy spectra, SHTOOLS also allows the computation of local energy spectra associated to specified sub-regions of the sphere.

Figure 10 shows the spherical harmonic energy spectrum computed from the vorticity field shown in Figure 8. There is a

power law range of approximately $k^{-2}$ extending over about a decade, from scales of about $25\,\mathrm{km}$ to $130\,\mathrm{km}$. This scaling is similar to the results for submesoscale turbulence from Soufflet et al. (2016), who find a slope of about -2 from scales of about $20\,\mathrm{km}$ to $80\,\mathrm{km}$ for a resolution $\Delta x = 2\,\mathrm{km}$.



**Figure 8.** Results of the baroclinic jet test case at 600 days near the surface at depth $z = -1.25$ m. The coarsest grid is $\Delta x_{\min} \approx 38$ km ($J = 5$) and the finest grid is $\Delta x_{\min} \approx 2.1$ km ($J = 9$), i.e. four levels of levels of local dyadic refinement. The tolerance is $\varepsilon = 0.02$. Top panel (left to right): adaptive grid, density perturbation, relative vorticity. Note that the green regions are the land mass, which are almost entirely at the coarsest level $J_{\min} = 5$, indicated by white in the leftmost figure. Bottom panel (left to right): adaptive grid, free surface perturbation, relative vorticity (note change in scale).



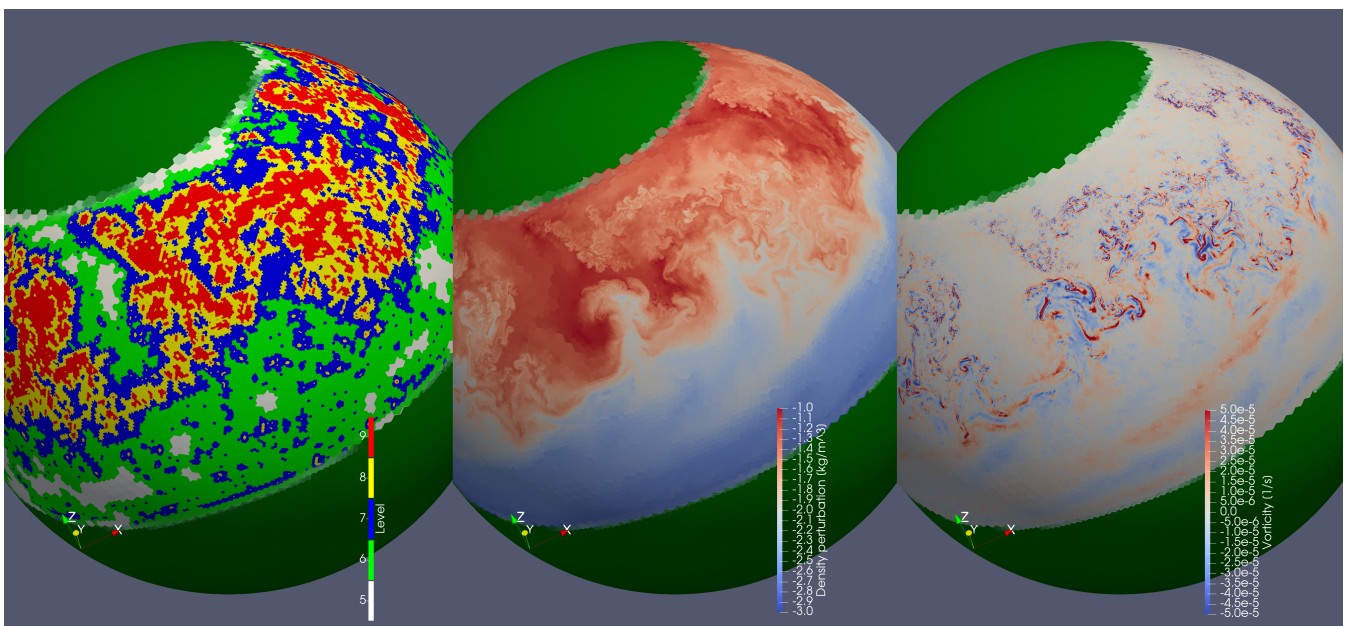

**Figure 9.** Higher compression run of the baroclinic jet test case with $\varepsilon = 0.06$ at 611 days. Adaptive grid (left), density perturbation (centre), relative vorticity (right). Compared with Figure 8 the grid is more localized, while still capturing the intense vorticity filaments.

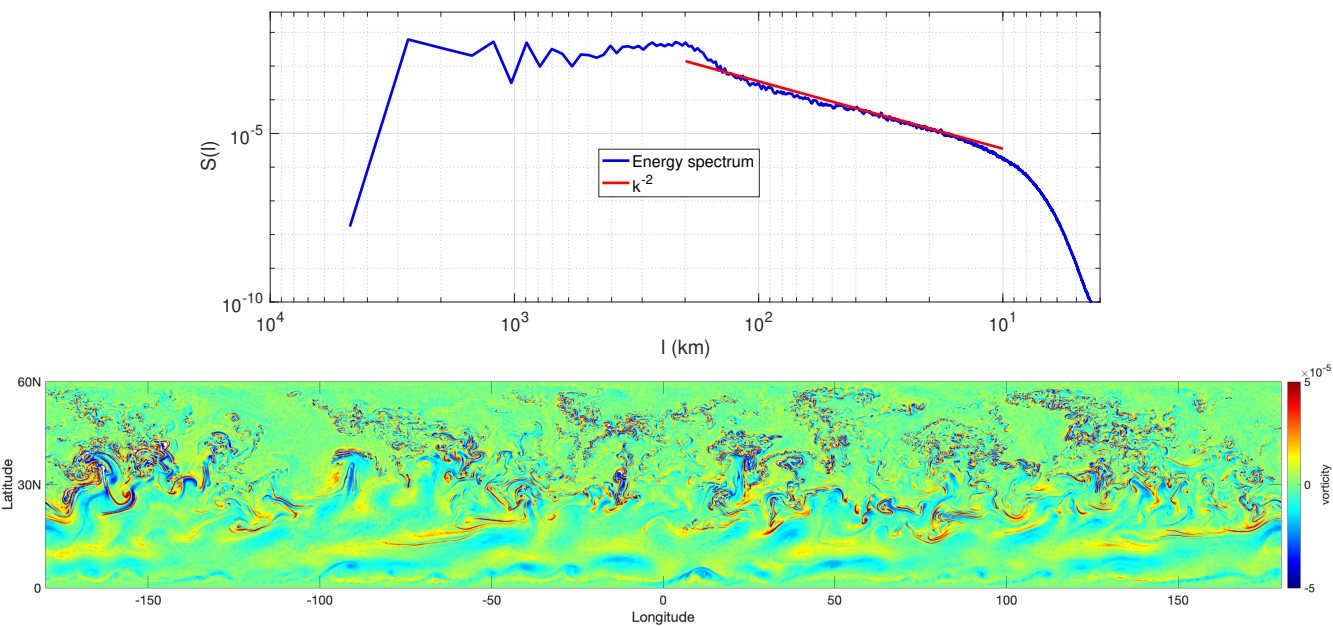

**Figure 10.** Top: spherical harmonics energy spectrum of baroclinic jet test case at 600 days with tolerance $\varepsilon = 0.02$ near the surface at depth $z = -1.25\,\mathrm{m}$ compared with $k^{-2}$ power law. Bottom: latitude-longitude projection of the associated vorticity field in the zonal channel with zonal length $6283\,\mathrm{km}$ (at the equator) and meridional width $1000\,\mathrm{km}$.



## 5 Conclusions

This paper introduced WAVETRISK-2.1, or WAVETRISK-OCEAN, the version of the dynamically adaptive code WAVETRISK
developed specifically for global ocean modelling. The dynamical equations of WAVETRISK-OCEAN are a multi-layer rotating
shallow water model with inhomogeneous density layers, but with no vertical variation of velocity and buoyancy within each
layer. This is an $n$-$\mathrm{IL}^0$ model in the terminology of Beron-Vera (2021). In such a model, to be consistent with the piecewise
constant representation of buoyancy in the vertical, a vertical average of the horizontal pressure gradient term in each layer is
used to compute horizontal velocity. For a seamount test case, we showed that the pressure gradient errors associated with this
discretization are of the same order of magnitude as those of state-of-the-art models based on a terrain-following coordinate.

Computationally, WAVETRISK-OCEAN uses the same wavelet-based adaptivity approach, hybrid tree–patch data structure
and `mpi` parallelization as WAVETRISK

The main new addition in WAVETRISK-OCEAN is the development of a semi-implicit barotropic–baroclinic mode splitting
time step. This relies on a simple and efficient adaptive multigrid elliptic solver, and is about 34–44 times faster than an
explicit scheme. WAVETRISK-OCEAN also includes conservative remapping using a piecewise parabolic scheme (PPR), vertical
diffusion with a turbulent kinetic energy (TKE) closure and volume penalization of horizontal solid boundaries.

We have verified the accuracy and performance of WAVETRISK-OCEAN on three standard test cases: seamount, upwelling
and unstable baroclinic jet. In the case of a complex flow such as the unstable baroclinic jet considered here, the adaptive
WAVETRISK-OCEAN model achieves an accuracy equivalent to that of non-adaptive models, but using significantly fewer grid
points.

WAVETRISK-OCEAN provides an innovative test bed for exploring the potential of dynamically adaptive methods for ocean
modelling. In particular, we are interested in using it to better understand the roles of barotropic and baroclinic dynamics in the
production and dissipation of turbulence.

Further development of WAVETRISK-OCEAN will include implementing volume penalization of bathymetry (in addition to
coastlines) (Debreu et al., 2020) and investigating more realistic configurations (e.g. with realistic coastline and bathymetry
geometry and external forcing).

*Code availability.* WAVETRISK-2.1 is published under the Creative Commons License 4.0 as https://doi.org/10.5281/zenodo.5608548.

*Author contributions.* NKRK prepared the manuscript with contributions from FL. NKRK developed the model code, with advice from FL
on appropriate approximations for ocean modelling. NKRK performed all model simulations using WAVETRISK-OCEAN. FL carried out the
CROCO simulations.

*Competing interests.* The authors declare that they have no conflict of interest.



*Acknowledgements.* NKRK acknowledges an NSERC Discovery grant and computer time from Compute Ontario (computeontario.ca) and Compute Canada (www.computecanada.ca). Alistair Adcroft provided valuable advice during the initial development of WAVETRISK-OCEAN. We are grateful to Thomas Dubos and Matthias Aechtner for their essential contributions to the WAVETRISK model on which

WAVETRISK-OCEAN is based.





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
