# Peer review of "WAVETRISK-2.1: an adaptive dynamical core for ocean modelling"

_Geoscientific Model Development, 2021_

## Referee Comment (RC1)

Review of

**WAVETRISK-2.1: an adaptive dynamical core for ocean modelling**

By N K-R Kevlahan and F Lemarie

This paper provides an overview of the ocean version of the WAVETRISK model (sections 1 and 2). It then describes, in section 3, a number of developments that have been made to the model specifically for the ocean; a split between barotropic and baroclinic steps; an elliptic (Helmholz) solver; a TKE vertical diffusion scheme. Section 3 also briefly describes the penalisation method for lateral boundaries and provides a helpful outline of the model's algorithm. Section 4 describes results obtained for 3 "standard" test cases.

The paper is well written and the developments described are impressive and well worth publication. I learnt a lot quite quickly reading it! I have made quite a number of comments that are mainly intended to help the authors to present their work accurately and more clearly. There is quite a body of literature here that was new to me. Several other readers will be in a similar position. So I hope my comments will help the authors to make their paper more accessible to such readers.

At the end of the review I have asked some questions about the dynamical formulation that the authors are exploring. Publication should not depend on the answers to these questions but it would be interesting to hear the authors views on these questions and perhaps some of these points could be mentioned in the concluding section as areas where detailed exploration would be valuable.

I haven't noted minor typos / grammatical errors as I am sure the journal editors will correct these.

**Abstract**

This is informative though the adaptation for the sphere is not mentioned and results from test cases are not summarised; it is difficult to summarise them briefly so I think that is OK.

Lines 10-12: innovative feature – is this mentioned in the main text? If not it could be mentioned in the introduction and/or conclusions.

WAVETRISK – it might be helpful to explain the origin of this name at some point. I think it is a shorthand for wavelet and TRisk but this is not very explicit.

**Introduction**

Lines 31-33: I think vertical columns are a good idea. It's not just a matter of simplifying development. But the authors might prefer to keep this statement as it is.

Line 34: "final stage in foundational set". This is a somewhat subjective statement. For example a non-linear equation of state might be viewed as part of the foundational set and isopycnal diffusion is also a very important component of ocean models.

Lines 39-40: Ripa (1993) introduced n-$IL^0$ for a reduced gravity ocean (no pressure variations at the bottom).

Line 42: Ripa emphasised that potential vorticity is not conserved (tracers are); it is advected with a source term. I think it is worth mentioning the Hamiltonian structure (as well as the Casimirs)

because this can be very helpful for developing discretisations with good conservation properties. There is a nice paper by Salmon illustrating that point.

Line 42-43: "ensures a good approximation of the horizontal pressure gradient". I don't think Ripa discussed that. Could you please provide a reference to support this statement? It might be better to discuss this point in section 4.1 when discussing the sea-mount test case.

Line 44: I think it is worth emphasising that Dubos et al (2015) derive their equations of motion from the discrete Hamiltonian somewhere near here.

Line 66-67: Are the details provided in section 3 new (not previously documented)? I think they are but this should be made explicit.

Section 2

Lines 72 – 83: The complexities of the equation of state are an important issue in ocean modelling. I think it is one of the main difficulties in the formulation of HLPEs. Postponing the treatment of these issues is understandable but will deter readers interested in practical applications. I wonder whether some of the issues for HLPEs might be less serious for ILPEs. A proper discussion of this point might well require a dedicated paper. A sentence or two referring to papers that discuss the issue for HLPEs might be a good "solution".

Line 89: is $v$ the vertex of a triangle?

Figure 1:   I do prefer figures which indicate how the hexagonal and triangular elements relate to each other. Just showing the triangular elements does not help readers unfamiliar with TRisk.

Equations (4) and (5): It would seem more natural to me to write $\overline{\theta_{ik}}^{e}$ than $\overline{\theta_{k}}^{e}$ but your equations appear to be consistent in dropping the $i$ subscript. It might be worth mentioning your convention as I found it quite puzzling at first.

Line 94: $q_{ek}$: I think potential vorticity is defined at the points marked by red triangles in Figure 1 rather than at the edges. I'm not sure what subscript is used to denote these vorticity points.

Line 95-96: $\overline{(.)}$ : the Coriolis term involves quantities at circulation and node points, so the inputs to the reconstruction in some cases are not just node quantities.

Lines 96-97: I think the discrete operators used are dimensionless. In other words strictly speaking they are difference operators. Calling them gradients or divergences is helpful as long as their difference operator nature is also made clear. This is important for (7) – (9) to be dimensionally correct.

Lines 99-100: This sentence when read together with the previous sentence is obscure. Does the additional N+1 vertical layer include depth integrated velocities v as well as the free surface height? If so, saying that would make the text much clearer.

Line 103: $\delta\theta_{k}$  : I think this should be $\delta_i\theta_{ik}$. The horizontal derivative must be within a layer (rather than horizontal in x,y,z space). It might be worth mentioning that.

 Lines 104-105: Ripa (1993) used vector invariant form (but also reduced gravity as mentioned earlier).

Equations (6) – (9): I haven't had time to check these. Unfortunately "horizontal" diffusion does not correspond to isopycnal diffusion.

Lines 119-120: "for better accuracy and stability" I imagine this separation does not need to be exact and is just intended to reduce the precision with which variables need to be stored. Is the mean independent of horizontal position?

Line 142: I found Dubos & Kevlahan (2013) to be a good starting point for understanding the wavelet scheme for the TRisk grid. It would be helpful to mention that.

Line 160: Typical internal gravity wave speeds are between 2 and 3 m/s not 0.05 m/s. I'm not sure how that error has crept into the text.

Lines 162 – 165: I mis-read this sentence at first despite being fully aware of what it is trying to say. I think the punctuation is difficult to follow. Please try to separate the 3 alternatives more clearly.

Line 173: "vortical": do you mean "baroclinic vertical modes" ??

Lines 175-176: for clarity please say explicitly that it would not represent barotropic tides accurately.

Lines 181-185: There is repetition of points about RK3 and RK4 that should be removed.

Lines 187-230: In general this is well explained, but I'm not completely clear about two things. First I think (please confirm) that (19) is solved on each of the three RK3 sub-steps. Second does the vertical diffusion use the fields on the backward time-step?  I think this is fairly clear but the text could be tidied up a little bit. More specifically, the point in the sentence on lines 188-189 could be moved to follow eqn (21).

Line 196: insert a comma after "position"

Lines 239-245 and Figure 2: is the Coriolis parameter equal to zero in these tests or are the results independent of it?

Lines 253-266 and Table 1: This would be easier to read if the sentence in lines 260-261 was brought forward to the start of line 255; I felt quite puzzled after reading lines 259 and Table 1. The computational core of ROMS has two tracers and might do some extra calculations (e.g. isopycnal diffusion) so the comparison is not very precise but it is useful and quite impressive in my view. The performance has only been shown for < 5 nodes. Are issues anticipated with configurations using 100 or more nodes?

Section 3.2: I did not notice any errors but I'm not very familiar with the TKE scheme.

Line 313: "baroclinic and vortical modes" again

Section 3.3: I was not previously aware of the SRJ method so found this section very interesting.

Section 3.5: I haven't understood how the adaptation of the horizontal grid differs from the wavelet transform algorithm (algorithm 4) which seems to involve adapting the points / zero wavelets. Algorithm 4 is called at a low level and hence very frequently whereas the adaptation of the horizontal grid is said to occur infrequently (once every 10 baroclinic steps if I recall correctly).

Section 4.1

I believe this seamount test case, as it has been set up, is an extremely rigorous test of this model in the sense that the horizontal inhomogeneity is an extreme case.

For this case is there a similar issue in your equations to that found in standard terrain following coordinates in the calculation of the hpg, i.e. that two large terms are nearly equal and opposite? I imagine there is. Some discussion of this point would be helpful to the reader.

I don't know what vertical discretisation you are using. Could you please direct the reader to the paper that describes this or provide at least a brief description of it.

It seems to me that this test case is less well defined / documented than it could be. I don't know how the results depend on the details of the vertical grid (number of levels and stretching). This complicates the comparisons.

SMcW (2003) use only 10 levels and the domain average kinetic energy of their best method appears to be smaller. Which figure in SMcW (2003) did you use to obtain the numbers you quote for their errors?

Beckmann & Haidvogel have a very different (actually quite strange) specification of the background density gradient and the density gradient that is included in the hpg calculation. It is quite difficult to compare your results with theirs.

Debreu et al 2018 do not say which algorithm they are using for the hpg calculation. I'm not sure what vertical grid spacing they use (is it uniform?). The results in your paper do look similar to those of Debreu et al 2108. Perhaps the simplest way to get a clean comparison would be to consult with Debreu and to use the same grid as that paper and give some more details of the scheme that he used.

When higher order reconstructions of fields are used to calculate the hpg, it becomes important to decide whether cell values represent point values or grid-cell means. The specification of the initial state is easier if the cell values are taken to be point values. If they are taken to be grid-cell means 3D volume integrals should be calculated to define the initial state. Could you please state what choice you have made here.

Equation after (29): S is the Burger number (not Burgers).

Line 358: I wonder whether removing the north-south boundaries has any impact on the results.

Figure 3 lower plot: The initial density field should be flat (horizontal isopycnals). So something is wrong in this figure.

Figure 4: It is interesting that there are no damped oscillations (unlike most low order schemes).

    Section 4.2

I am not familiar with this test case so cannot offer detailed comments on it.

Line 385: I wonder what the value of \beta is in the centre of this domain. Is the impact of the variable Coriolis parameter on the results negligible?

Table 3: layers 3 and 4 have typos in the decimal place for z and dz.

Lines 395-397: The integration J8 that runs with $r_{max}$ = 0.66 is impressive. I think I have not grasped the point about this only being possible because the grid is adaptive. The text says that J8 is a non-adaptive integration.

Lines 407-408. The differences between these simulations and CROCO are very large. I am not sure what to make of this as a test case.

Figure 6: I suppose that most readers will look at the non-adaptive simulation J8 and compare it with J6J8 to see how closely they compare. J6J8 vertical velocity has more small-scale structure than J8. I think that could be a cause for concern.

Section 4.3

Line 432: Again I wonder what \beta is for this configuration and whether that is relevant to the results.

Figures 8 and 9: There is not much discussion of these. I wonder whether it would be more helpful to present the fields in figure 9 for the smaller & larger tolerances in a single figure. If the fields are "directly" comparable that could be more informative.

Figure 10: Upper panel: Would it be interesting to present results below the surface where $k^{-3}$ power spectrum might be expected?

Figure 10: Lower panel: It is hard to see the detail. Perhaps limiting the section to 100 degrees of longitude would enable the plot to be made larger and the detail to be more visible.

Conclusions

Lines 479-480: this conclusion may well be true (which would be exciting) but the test case is not as standardised as it could be.

Lines 488-489: I didn't pick this up whilst reading section 4.3. Could this be explained more explicitly there?

Lines 491-493: It is also a good test-bed for exploring some important questions about the strengths and weaknesses of the numerical formulations (see the following points).

Some questions about the basic formulations

A well-known difficulty with the TRisk formulation relates to the ratio of the number of degrees of freedom of its velocity and tracer fields. This affects the number of inertia-gravity waves supported by the discretisation. Some of the "additional" inertia-gravity waves can generate computational modes (see e.g. Cotter & Shipton https://doi.org/10.1016/j.jcp.2012.05.020). Is this a serious problem and how is it controlled in the author's model?

Another issue relates to the accuracy/consistency of some of the terms: Peixoto (2016) https://doi.org/10.1016/j.jcp.2015.12.058 discusses this issue and there are more recent papers following up on this point.

Level PE models using the vector invariant form of the momentum equations can suffer from the Hollingsworth instability whilst it seems that homogeneous layer (HLPE) models do not. I wonder whether your ILPEs are similarly not prone to this problem and whether the calculations in Bell et al (2016) **https://doi.org/10.1002/qj.2950** which provide an explanation of the difference between the layer and level models could be repeated using the ILPE equations.

As I understand it homogeneous layers are first order accurate approximations to a continuously stratified fluid. But if the calculations are "centred" they are nonetheless second order accurate. In traditional ocean models (like NEMO) it is not entirely clear whether tracer values represent cell means or spot values. In your layer model this is more explicit. Is your model second order (or more) accurate? Is there any attempt to calculate the hpg using higher order reconstructions of the fields similar to those of Shchepetkin & McWilliams (2003)?

---

## Author Comment (AC1)

**Response to Review Mike Bell's Comments (RC1)**
We would like to thank Mike for his careful reading of the paper and his extremely detailed and helpful comments. They have enabled us to clarify the presentation and to address several important points we had not considered previously.

We have responded to each comment below. The Reviewer's comments are in **black text** and our responses are in **blue text**.

Review of

WAVETRISK-2.1: an adaptive dynamical core for ocean modelling

By N K-R Kevlahan and F Lemarie

This paper provides an overview of the ocean version of the WAVETRISK model (sections 1 and 2). It then describes, in section 3, a number of developments that have been made to the model specifically for the ocean; a split between barotropic and baroclinic steps; an elliptic (Helmholz) solver; a TKE vertical diffusion scheme. Section 3 also briefly describes the penalisation method for lateral boundaries and provides a helpful outline of the model's algorithm. Section 4 describes results obtained for 3 "standard" test cases.

The paper is well written and the developments described are impressive and well worth publication. I learnt a lot quite quickly reading it! I have made quite a number of comments that are mainly intended to help the authors to present their work accurately and more clearly. There is quite a body of literature here that was new to me. Several other readers will be in a similar position. So I hope my comments will help the authors to make their paper more accessible to such readers.

At the end of the review I have asked some questions about the dynamical formulation that the authors are exploring. Publication should not depend on the answers to these questions but it would be interesting to hear the authors views on these questions and perhaps some of these points could be mentioned in the concluding section as areas where detailed exploration would be valuable.

I haven't noted minor typos / grammatical errors as I am sure the journal editors will correct these.

Abstract

This is informative though the adaptation for the sphere is not mentioned and results from test cases are not summarised; it is difficult to summarise them briefly so I think that is OK.

Thank you: we will not add more information about the test cases.

Lines 10-12: innovative feature – is this mentioned in the main text? If not it could be mentioned in the introduction and/or conclusions.

We have added a comment that WAVETRISK could be used to build a coupled ocean-atmosphere model to the conclusions.

WAVETRISK – it might be helpful to explain the origin of this name at some point. I think it is a short- hand for wavelet and TRisk but this is not very explicit.

We have included an explanation of the name WAVETRISK in the introduction.

Introduction

Lines 31-33: I think vertical columns are a good idea. It's not just a matter of simplifying development. But the authors might prefer to keep this statement as it is.

We have included "better accuracy" as an additional reason to use vertical columns.

Line 34: "final stage in foundational set". This is a somewhat subjective statement. For example a non-linear equation of state might be viewed as part of the foundational set and isopycnal diffusion is also a very important component of ocean models.

We have changed "final stage" to "significant step". We have implemented a nonlinear equation of state in the code, but have not used it yet.

Lines 39-40: Ripa (1993) introduced n-IL0 for a reduced gravity ocean (no pressure variations at the bottom).

We have added this detail.

Line 42: Ripa emphasised that potential vorticity is not conserved (tracers are); it is advected with a source term. I think it is worth mentioning the Hamiltonian structure (as well as the Casimirs) because this can be very helpful for developing discretisations with good conservation properties. There is a nice paper by Salmon illustrating that point.

These changes have been made.

Line 42-43: "ensures a good approximation of the horizontal pressure gradient". I don't think Ripa discussed that. Could you please provide a reference to support this statement? It might be better to discuss this point in section 4.1 when discussing the seamount test case.

This is indeed a good remark. The discussion about the horizontal pressure gradient has now been moved completely to section 4.1 and this particular sentence has been removed from the introduction. It turns out that there was an error in the results presented in section 4 for the seamount test case: the viscosity used was much higher than that of the studies with which we compared the results. With the appropriate lower viscosity, we get pressure gradient errors typical of a sigma coordinate model. As you had foreseen, the Ripa model does not reduce this type of error.

We have highlighted improving the TRiSK approximation of the horizontal pressure gradient as a development priority in the conclusions.

Line 44: I think it is worth emphasising that Dubos et al (2015) derive their equations of motion from the discrete Hamiltonian somewhere near here.

Added to the previous paragraph.

Line 66-67: Are the details provided in section 3 new (not previously documented)? I think they are but this should be made explicit.

We now indicate that these details are presented here for the first time.

Section 2

Lines 72 – 83: The complexities of the equation of state are an important issue in ocean modelling. I think it is one of the main difficulties in the formulation of HLPEs. Postponing the treatment of these issues is understandable but will deter readers interested in practical applications. I wonder whether some of the issues for HLPEs might be less serious for ILPEs. A proper discussion of this point might well require a dedicated paper. A sentence or two referring to papers that discuss the issue for HLPEs might be a good "solution".

In the HLPEs density in each layer is assumed constant in space and time while the only assumption in the ILPEs is to consider state variables piecewise constant in the vertical. Unlike the HLPEs, with ILPEs density is not used as a regridding variable (i.e. layers are not defined using target interface densities) and the formulation appears close to a standard vertical ALE algorithm. However, a common issue that must be tackled is the additional source of pressure gradient error stemming from the use of a nonlinear equation of state including compressibility of seawater (Dukowicz, 2001).

Line 89: is v the vertex of a triangle?

No: v labels a triangular cell. We now specify this.

Figure 1: I do prefer figures which indicate how the hexagonal and triangular elements relate to each other. Just showing the triangular elements does not help readers unfamiliar with TRisk.

 We have added a diagram showing the relation of the primal and dual grids.

Equations (4) and (5): It would seem more natural to me to write  than  but your equations appear to be consistent in dropping the i subscript. It might be worth mentioning your convention as I found it quite puzzling at first.

We agree that dropping the horizontal subscripts is a bit confusing, and so we have added them back.

Line 94: : I think potential vorticity is defined at the points marked by red triangles in Figure 1 rather than at the edges. I'm not sure what subscript is used to denote these vorticity points.

We now explain that this term uses potential vorticity at v points (triangles) reconstructed at e points (edges) using the TRiSK formula.

Line 95-96: (. ) : the Coriolis term involves quantities at circulation and node points, so the inputs to the reconstruction in some cases are not just node quantities.

Please see the previous response. The TRiSK approximation of the perpendicular flux of potential vorticity involves a fairly complex discretization involving quantities at triangle, edge and node points.

Lines 96-97: I think the discrete operators used are dimensionless. In other words strictly speaking they are difference operators. Calling them gradients or divergences is helpful as long as their difference operator nature is also made clear. This is important for (7) – (9) to be dimensionally correct.

For simplicity, these operators are defined to be the complete discrete differential operators, including the appropriate edge lengths and hexagon areas. The notation used for the horizontal dissipation operators (7-9) was not consistent and this has been corrected.

Lines 99-100: This sentence when read together with the previous sentence is obscure. Does the additional N+1 vertical layer include depth integrated velocities v as well as the free surface height? If so, saying that would make the text much clearer.

We now state explicitly that we do not store the depth integrated velocities, since they are not required for the semi-implicit barotropic-baroclinic algorithm.

Line 103: $\delta \theta_k$: I think this should be $\delta_i \theta_{ik}$. The horizontal derivative must be within a layer (rather than horizontal in x,y,z space). It might be worth mentioning that.

Corrected.

Lines 104-105: Ripa (1993) used vector invariant form (but also reduced gravity as mentioned earlier).

Corrected.

Equations (6) – (9): I haven't had time to check these. Unfortunately "horizontal" diffusion does not correspond to isopycnal diffusion.

Indeed in the ILPEs case along-layer diffusion does not correspond to isopycnal diffusion. For the present paper, equations are formulated with along-layer diffusion, but a rotation tensor could be applied to orientate it differently. Note that in the case of a linear EOS, an isopycnal diffusion operator would be identically null. We have replaced "horizontal diffusion" by "along-layer diffusion" throughout the paper.

Lines 119-120: "for better accuracy and stability" I imagine this separation does not need to be exact and is just intended to reduce the precision with which variables need to be stored. Is the mean independent of horizontal position?

The idea is to separate the densities into a mean component (that is constant in time) and a fluctuating component. This is helpful in cases where a mean value should be exactly conserved. The mean value need not be independent of horizontal position, just time-independent.

Line 142: I found Dubos & Kevlahan (2013) to be a good starting point for understanding the wavelet scheme for the TRisk grid. It would be helpful to mention that.

We have added this reference.

Line 160: Typical internal gravity wave speeds are between 2 and 3 m/s not 0.05 m/s. I'm not sure how that error has crept into the text.

Thanks for pointing this out! We have corrected this typo.

Lines 162 – 165: I mis-read this sentence at first despite being fully aware of what it is trying to say. I think the punctuation is difficult to follow. Please try to separate the 3 alternatives more clearly.

The presentation has been clarified.

Line 173: "vortical": do you mean "baroclinic vertical modes" ??

We meant "baroclinic vertical modes" and "horizontal geostrophic (vortical) modes" (e.g. 2D turbulence). We have re-written this sentence.

Lines 175-176: for clarity please say explicitly that it would not represent barotropic tides accurately. Lines 181-185: There is repetition of points about RK3 and RK4 that should be removed.

We have added the fact that it would not represent barotropic tides accurately. We have deleted these repeated points.

Lines 187-230: In general this is well explained, but I'm not completely clear about two things. First I think (please confirm) that (19) is solved on each of the three RK3 sub-steps. Second does the vertical diffusion use the fields on the backward time-step? I think this is fairly clear but the text could be tidied up a little bit. More specifically, the point in the sentence on lines 188-189 could be moved to follow eqn (21).

No: as detailed in Algorithm 1, the elliptic equation (19) is solved once after advancing completing the explicit RK3 step.

As mentioned in 188-189 and in Algorithm 1, vertical diffusion is implemented in a separate backwards Euler split step.

We have moved the sentence on lines 188-189 after (21).

Line 196: insert a comma after "position"

Done.

Lines 239-245 and Figure 2: is the Coriolis parameter equal to zero in these tests or are the results independent of it?

The Coriolis parameter f = 1e-4 1/s and dt = 360 s; the results depend on these parameters only via their product, dt * f. We have added this information to the figure caption.

Lines 253-266 and Table 1: This would be easier to read if the sentence in lines 260-261 was brought forward to the start of line 255; I felt quite puzzled after reading lines 259 and Table 1. The computational core of ROMS has two tracers and might do some extra calculations (e.g. isopycnal diffusion) so the comparison is not very precise but it is useful and quite impressive in my view. The performance has only been shown for < 5 nodes. Are issues anticipated with configurations using 100 or more nodes?

The sentence has been moved as suggested.

We have added a sentence explaining that the comparison is not exact, since ROMS may do extra operations.

We have run wavetrisk on up to 64 nodes (2400 cores) on the Compute Canada machine without issue. The domain decomposition technique we use means that the maximum number of cores is determined by the coarsest grid resolution and the patch size. For example, if the coarsest grid is J = 7 (seven bisections of the icosahedron) and the patch size is 4x4 then we can use at most 2560 cores.

Section 3.2: I did not notice any errors but I'm not very familiar with the TKE scheme.

Noted.

Line 313: "baroclinic and vortical modes" again

Corrected.

Section 3.3: I was not previously aware of the SRJ method so found this section very interesting.

We are glad you found it interesting! It is quite a clever technique to accelerate Jacobi iterations.

Section 3.5: I haven't understood how the adaptation of the horizontal grid differs from the wavelet transform algorithm (algorithm 4) which seems to involve adapting the points / zero wavelets. Algorithm 4 is called at a low level and hence very frequently whereas the adaptation of the horizontal grid is said to occur infrequently (once every 10 baroclinic steps if I recall correctly).

The wavelet transform Algorithm 4 (which is called after each sub-step) transforms the solution back onto the entire grid (active zone, adjacent zone and grid points required for TRiSK stencils). This ensures the error satisfies the relative tolerance epsilon over the whole grid, conserves mass and that minimal energy is lost. It is required since additional grid points are included to be able to compute the TRiSK operators (with values supplied by interpolation), but they are not in the active grid. Grid adaptation involves modifying the active grid based on the new solution by testing wavelets of all prognostic variables at edges and nodes. This is explained in Kevlahan and Dubos (2019). We have modified the title of Algorithm 4 to make this clearer.

Section 4.1

I believe this seamount test case, as it has been set up, is an extremely rigorous test of this model in the sense that the horizontal inhomogeneity is an extreme case.

For this case is there a similar issue in your equations to that found in standard terrain following coordinates in the calculation of the hpg, i.e. that two large terms are nearly equal and opposite? I imagine there is. Some discussion of this point would be helpful to the reader.

You are completely correct. The horizontal pressure gradient is composed of two terms (see eq. 5) : an along-layer hydrostatic pressure term in the Bernoulli function and one related to the along-layer geopotential gradient. The issue is indeed similar to the one found in standard terrain-following coordinates or, more generally, in coordinate systems non-aligned with geopotential surfaces. This simple approximation is inherited from DYNAMICO and should be improved.

I don't know what vertical discretisation you are using. Could you please direct the reader to the paper that describes this or provide at least a brief description of it.

As we point out on p 16 (new version), the seamount test case uses a Chebyshev hybrid vertical grid with a Lorenz placement of variables (as in DYNAMICO). We have modified the description of the vertical layers in Section 2.1 to make this clear, as well as defining the grid on p 16.

It seems to me that this test case is less well defined / documented than it could be. I don't know how the results depend on the details of the vertical grid (number of levels and stretching). This complicates the comparisons.

We are interested in quantifying the horizontal pressure gradient errors associated with the Ripa scheme. With a small viscosity, we find that at large Burger numbers they are

much larger than those of the other models we compare with. The comparisons with other results are intended only to give a general idea of how well our model performs.

SMcW (2003) use only 10 levels and the domain average kinetic energy of their best method appears to be smaller. Which figure in SMcW (2003) did you use to obtain the numbers you quote for their errors?

We point out that they use only 10 levels. We obtained the comparison numbers from their figure 5 (top right). We now indicate this in the paper.

Beckmann & Haidvogel have a very different (actually quite strange) specification of the background density gradient and the density gradient that is included in the hpg calculation. It is quite difficult to compare your results with theirs.

We agree, and we now point this out.

Debreu et al 2018 do not say which algorithm they are using for the hpg calculation. I'm not sure what vertical grid spacing they use (is it uniform?). The results in your paper do look similar to those of Debreu et al 2018. Perhaps the simplest way to get a clean comparison would be to consult with Debreu and to use the same grid as that paper and give some more details of the scheme that he used.

As mentioned earlier, with the corrected (much smaller) viscosity the HPG errors are now much larger than those of Debreu et al. (2018). Croco the default scheme for HPG calculation is the Density-Jacobian scheme using cubic polynomial fits described in SMcW (2003). We therefore now conclude that we should investigate using improved discretization of the HPG.

When higher order reconstructions of fields are used to calculate the hpg, it becomes important to decide whether cell values represent point values or grid-cell means. The specification of the initial state is easier if the cell values are taken to be point values. If they are taken to be grid-cell means 3D volume integrals should be calculated to define the initial state. Could you please state what choice you have made here.

WAVETRISK is derived as a consistent second-order method and we interpret cell values as grid-cell means. We do not use higher order reconstructions of fields.

Equation after (29): S is the Burger number (not Burgers).

Corrected.

Line 358: I wonder whether removing the north-south boundaries has any impact on the results.

There are no north-south boundaries, but the results are relatively insensitive to the radius of the planet.

Figure 3 lower plot: The initial density field should be flat (horizontal isopycnals). So something is wrong in this figure.

Thanks for catching this (we accidentally left on a zonal average switch in the visualization). We double-checked the density initialization and corrected the figure.

Figure 4: It is interesting that there are no damped oscillations (unlike most low order schemes). Section 4.2

I am not familiar with this test case so cannot offer detailed comments on it.

Noted.

Line 385: I wonder what the value of \beta is in the centre of this domain. Is the impact of the variable Coriolis parameter on the results negligible?

Beta is approximately 7e-10 in the centre of the domain. We did not notice any impact of variable Coriolis.

Table 3: layers 3 and 4 have typos in the decimal place for z and dz.

Corrected.

Lines 395-397: The integration J8 that runs with r_{max} = 0.66 is impressive. I think I have not grasped the point about this only being possible because the grid is adaptive. The text says that J8 is a non-adaptive integration.

J8 is indeed a non-adaptive simulation.  It is the adaptive simulation that runs with r_max = 0.66 at its coarsest resolution J6. We have added text to make this clear.

Lines 407-408. The differences between these simulations and CROCO are very large. I am not sure what to make of this as a test case.

Although our setup is necessarily different, due to the spherical geometry of our global model (e.g. we use a zonal channel and are not on a beta-place), we consider that our results are qualitatively similar to the CROCO results. Note also that the CROCO results have been updated to use the same axes as the WAVETRISK results.

Figure 6: I suppose that most readers will look at the non-adaptive simulation J8 and compare it with J6J8 to see how closely they compare. J6J8 vertical velocity has more small-scale structure than J8. I think that could be a cause for concern.

The other simulations were run without any along-level diffusion. Some numerical diffusion is normally required for centred schemes, so we have re-run the low resolution J6J8 case with a small viscosity 5.5 m^2/s and the vertical velocity looks smoother.

Section 4.3

Line 432: Again I wonder what \beta is for this configuration and whether that is relevant to the results.

We have added the information that the rotation rate Omega = 1e-4 1/s to match the Coriolis parameter in Soufflet (2016), corresponding to to \beta = 1.73e-10 in the centre of the channel. We also considered forcing \beta to match Soufflet's value \beta = 1.6e-11 in the centre of the channel at 30N, but this would require an extremely large planetary radius, about 10 000 km. Our results are therefore not intended to be an exact comparison, but demonstrate the ability of our code to simulate an unstable, turbulent baroclinic jet.

Figures 8 and 9: There is not much discussion of these. I wonder whether it would be more helpful to present the fields in figure 9 for the smaller & larger tolerances in a single figure. If the fields are "directly" comparable that could be more informative.

We think it is helpful to include large figures for the results, since seeing the structure of the small scale features is important.  We have added an additional sentence to the discussion of these figures.

Figure 10: Upper panel: Would it be interesting to present results below the surface where k^{-3} power spectrum might be expected?

Thanks for this suggestion. We now compare the energy spectrum in the top layer (z = -1.25 m) with the energy spectrum at z = -887 m. The energy spectrum at depth is indeed close to k^{-3} (slightly shallower), typical of a forward cascade of enstrophy in barotropic flow, while the k^{-2} energy spectrum at the surface is typical of baroclinic submesoscale turbulence. The transition between the two types of turbulence occurs between z = -642 m and -283 m.

Figure 10: Lower panel: It is hard to see the detail. Perhaps limiting the section to 100 degrees of longitude would enable the plot to be made larger and the detail to be more visible.

We agree that it is hard to see the detail at normal resolution. However, this is a high resolution figure so the reader can zoom in to see the details (GMD is only published on online, so lack of resolution of a print copy is not a problem). We think it is helpful to show the full periodic domain.

Conclusions

Lines 479-480: this conclusion may well be true (which would be exciting) but the test case is not as standardised as it could be.

We have added a note of caution.

Lines 488-489: I didn't pick this up whilst reading section 4.3. Could this be explained more explicitly there?

We have added an introductory section to the beginning of section 4, motivating the three test cases and explaining what sort of verification we are doing, and its limits. We have modified this sentence in the conclusions, removing the term "accuracy", to make it clear that the adaptive results are qualitatively reasonable.

Lines 491-493: It is also a good test-bed for exploring some important questions about the strengths and weaknesses of the numerical formulations (see the following points).

Some questions about the basic formulations

A well-known difficulty with the TRisk formulation relates to the ratio of the number of degrees of freedom of its velocity and tracer fields. This affects the number of inertia-gravity waves supported by the discretisation. Some of the "additional" inertia-gravity waves can generate computational modes (see e.g. Cotter & Shipton https://doi.org/10.1016/j.jcp.2012.05.020). Is this a serious problem and how is it controlled in the author's model?

We have not seen issues with additional computational modes in wavetrisk, even with fully explicit time stepping. It is possible that this is because wavetrisk is a global model on the sphere where the primal and dual grids are composed of irregular triangles and hexagons and the beta parameter is not constant. Note also that the barotropic-baroclinic mode splitting used here damps the inertia-gravity waves to some extent.

Another issue relates to the accuracy/consistency of some of the terms: Peixoto (2016)

https://doi.org/10.1016/j.jcp.2015.12.058 discusses this issue and there are more recent papers following up on this point.

One of us (NK) did discuss this issue with Peixoto, and we implemented there scheme. However, in practice we found it to be less stable that the TRiSK formulation.

Level PE models using the vector invariant form of the momentum equations can suffer from the Hollingsworth instability whilst it seems that homogeneous layer (HLPE) models do not. I wonder whether your ILPEs are similarly not prone to this problem and whether the calculations in Bell et al (2016) https://doi.org/10.1002/qj.2950 which provide an explanation of the difference between the layer and level models could be repeated using the ILPE equations.

It would be very interesting to extend the calculations of Bell et al (2016) to the case of ILPEs, but this seems to us to go too far beyond the objectives of the present paper. However, Hollingsworth instability does not seem to occur in the WAVETRISK and DYNAMICO numerical simulations using the optimized icosahedral C-grid on the sphere.

As I understand it homogeneous layers are first order accurate approximations to a continuously stratified fluid. But if the calculations are "centred" they are nonetheless second order accurate. In traditional ocean models (like NEMO) it is not entirely clear whether tracer values represent cell means or spot values. In your layer model this is more explicit. Is your model second order (or more) accurate? Is there any attempt to calculate the hpg using higher order reconstructions of the fields similar to those of Shchepetkin & McWilliams (2003)?

Like DYNAMICO, on the plane, our model is uniformly second-order accurate, but on the sphere the error drops locally near the pentagon points. In our case, tracer values are cell means, although cell means and spot values should be equivalent up to second order accuracy. We have not attempted to use higher order reconstructions of the fields.

---

## Author Comment (AC2)

**Response to Anonymous Referee Comments (RC2)**

We would like to thank the Referee for reading our paper. We reply to their comments below.

This is a well-written paper on the ocean version of the WAVETRISK model worth publication. It discusses the equations being solved along with their spatial discretizations, the time-stepping module, the vertical diffusion and TKE closure, the penalisation of lateral boundaries, outline of the relevant algorithms, and finally numerical results against some standard test cases.

I was able to view the very detailed review of Referee # 1. I agree with his recommendations and suggestions. He has addressed every relevant issue worth mentioning.

We have replied to each of Referee 1's many comments in detail and made many modifications to address his comments and improve the paper. These edits are indicated in the revised version of the manuscript.

My only personal recommendation would be to add some convergence plots (to test the accuracy of the WAVETRISK model) and some scaling plots e.g. against the number of cores (to test the performance).

Since the test cases do not have exact solutions, it is not possible to present true convergence plots. Nevertheless, Section 4.3 confirms that the results for the upwelling test case are qualitatively similar when the resolution is increased. The convergence and error control properties of the adaptive WAVETRISK algorithm have already been extensively verified and quantified in our previous papers, for example in figures 14-17 (Dubos and Kevlahan 2013), figures 10-13 (Aechtner, Kevlahan, Dubos 2015), figure 5 (Kevlahan, Dubos, Aechtner 2015). The current barotropic-baroclinic splitting inherits these basic error control properties.

Similarly, Kevlahan and Dubos (2019) included a detailed evaluation of the parallel performance of the adaptive and parallelized WAVETRISK algorithm (see figure 4 and table 2). The current paper states the overhead associated with the splitting (3-30%), but the parallel scaling is not affected. In table 1 of the current paper we complement the previous scaling results with an evaluation of the computational performance of wavetrisk-ocean run non-adaptively compared with ROMS. This assesses the basic efficiency of the code.

---

## Referee Report (RR1)

Review of Manuscript 3 of WAVETRISK-2.1: an adaptive dynamical core for ocean modelling

By N K-R Kevlahan and F Lemarie

The authors have given clear responses to nearly all my comments and modified their paper accordingly. I recommend that the paper is accepted subject to one revision and two minor suggestions that the authors can accept or reject as they see fit.

I note that I found it quite difficult to check the changes made in response to my comments because the authors did not give me the lines in the new version of the paper corresponding to the comments. Also I didn't see a list of the changes that have been made to the paper despite the fact that some of the changes (for example to the results in section 4.1) are quite substantial.

**Requested revision**

Lines 525-527. The first sentence summarising the results for the sea mount test case has not been revised in line with the changes to the results.  This sentence should be revised. The "note of caution" that has been added in the following sentence is not a sufficient qualification. The earlier text is quite clear that the present hpg scheme needs to be improved (lines 419-420).

**Suggestions for Minor revisions**

1. Sentence following (7)-(9): The authors note in their responses that the along-layer diffusion would be zero for the case of a linear EOS. I had not realised this and think that it would be worth pointing it out explicitly. I think it relies on the layers being initialised with uniform potential temperatures and retaining those values.
2. Lines 264 and 271: The authors moved a sentence to line 264 following my comment. I actually meant to suggest that the sentence now on line 271 (starting "Since it uses …") be moved to line 264 (after " $\tau \approx 1 \micro s$")!

---

## Author Response (AR2)

**Response to Referee 1 (Mike Bell)'s Comments on the Revised Manuscript**

We have responded to each comment below. The Reviewer's comments are in **black text** and our responses are in **blue text**.

The authors have given clear responses to nearly all my comments and modified their paper accordingly. I recommend that the paper is accepted subject to one revision and two minor suggestions that the authors can accept or reject as they see fit.

I note that I found it quite difficult to check the changes made in response to my comments because the authors did not give me the lines in the new version of the paper corresponding to the comments. Also I didn't see a list of the changes that have been made to the paper despite the fact that some of the changes (for example to the results in section 4.1) are quite substantial.

We thought that it would be sufficient to describe the changes in the responses and highlight them in the text. However, we agree that because there have been extensive revisions we should really have included the line numbers of the changes from the revised version. To simplify the presentation we highlighted all changes made as direct responses to referee comments (consistent with the style guidelines to highlight "relevant" changes).

**Requested revision**
Lines 525-527. The first sentence summarising the results for the sea mount test case has not been revised in line with the changes to the results. This sentence should be revised. The "note of caution" that has been added in the following sentence is not a sufficient qualification. The earlier text is quite clear that the present hpg scheme needs to be improved (lines 419-420).

Thanks for catching this: we missed updating this sentence in the conclusions and have now modified it at lines 527 to 531 (track changes copy) to properly reflect the new results.

**Suggestions for Minor revisions**
1. Sentence following (7)-(9): The authors note in their responses that the along-layer diffusion would be zero for the case of a linear EOS. I had not realised this and think that it would be worth pointing it out explicitly. I think it relies on the layers being initialised with uniform potential temperatures and retaining those values.

As we stated in our response, in the current WAVETRISK implementation it is diffusion *along an isopycnal* that is exactly zero and therefore does not make sense (in the case of a single scalar and linear EOS). The implemented along-layer diffusion is indeed non-zero for inhomogeneous density layers.

2. Lines 264 and 271: The authors moved a sentence to line 264 following my comment. I actually meant to suggest that the sentence now on line 271 (starting "Since it uses ...") be moved to line 264 (after " $\tau \approx 1 \micro s$")!

Thank you for the clarification: we have now modified the text at lines 264 to 273 (track changes copy) as you had originally intended.